# A suppression-modification gene drive for malaria control targeting the ultra-conserved RNA gene mir-184

Sebald A. N. Verkuijl [1,4], Giuseppe Del Corsano [1,4], Paolo Capriotti [1,4], Pei-Shi Yen[1], Maria Grazia Inghilterra[1], Prashanth Selvaraj [2], Astrid Hoermann [1], Aida Martinez-Sanchez[3], Chiamaka Valerie Ukegbu[1], Temesgen M. Kebede[1], Dina Vlachou [1], George K. Christophides [1] & Nikolai Windbichler [1] ✉

Gene drive technology presents a promising approach to controlling malaria vector populations. Suppression drives are intended to disrupt essential mosquito genes whereas modification drives aim to reduce the individual vectorial capacity of mosquitoes. Here we present a highly efficient homing gene drive in the African malaria vector *Anopheles gambiae* that targets the microRNA gene mir-184 and combines suppression with modification. Homozygous gene drive (miR-184^D) individuals incur significant fitness costs, including high mortality following a blood meal, that curtail their propensity for malaria transmission. We attribute this to a role of miR-184 in regulating solute transport in the mosquito gut. However, females remain fully fertile, and pure-breeding miR-184^D populations suitable for large-scale releases can be reared under laboratory conditions. Cage invasion experiments show that miR-184^D can spread to fixation thereby reducing population fitness, while being able to propagate a separate antimalarial effector gene at the same time. Modelling indicates that the miR-184^D drive integrates aspects of population suppression and population replacement strategies into a candidate strain that should be evaluated further as a tool for malaria eradication.

The fight against malaria has begun to stall in the last decade, and no net reduction in cases, of which there were 249 million globally in 2022, has been achieved in recent years[1]. Despite the rollout of the RTS,S/AS01 malaria vaccine and the availability of dual-ingredient insecticide-treated bed nets the world is not currently on track to achieve the milestones laid out in the 2016-2030 Global Technical Strategy for malaria[2]. The rise of urban malaria vectors is a further challenge in addition to widespread drug, diagnostic and insecticide resistance in Africa[1]. This highlights the need for novel tools and

strategies to control the disease, especially if malaria elimination is to remain a viable prospect.

Homing gene drives were originally proposed 20 years ago[3] as a potential malaria vector control tool and it was subsequently demonstrated that the homing mechanism efficiently biases inheritance in *Anopheles* mosquitoes[4]. The advent of CRISPR endonucleases then provided the flexibility to rapidly conceive and test a range of novel gene drive designs in malaria vectors[5–10]. Suppressive gene drives are intended to decrease malaria vector population size by targeting genes

[1]Department of Life Sciences, Faculty of Natural Sciences, Imperial College London, London, UK. [2]Institute for Disease Modeling, Bill and Melinda Gates Foundation, Seattle, WA, USA. [3]Section of Cell Biology and Functional Genomics, Department of Metabolism, Digestion and Reproduction, Faculty of Medicine, Imperial College London, London, UK. [4]These authors contributed equally: Sebald A. N. Verkuijl, Giuseppe Del Corsano, Paolo Capriotti. ✉e-mail: n.windbichler@imperial.ac.uk

essential for mosquito fitness. Strategies that disrupt female development or skew the sex ratio towards males are among the most promising approaches being explored[9–14]. Alternatively, homing gene drives aimed at population modification are designed to reduce the individual vectorial capacity of mosquito vectors by propagating effector traits[5,8,15–18] that interfere with the development of the *plasmodium* parasite.

These two strategies have unique features and benefits. On the one hand, suppression drives target critical and conserved mosquito genes to eliminate vector populations, sharing the same goal as many other current vector control tools (e.g. breeding site elimination). However, gene drive population suppression causes the strong selection for gene drive resistance and may result in empty breeding sites or niches that could be reinvaded by wild-type vectors of the same or possibly a different species. Modification drives, on the other hand, sidestep perceived environmental concerns surrounding population suppression as well as logistical issues (e.g. the need for constant backcrossing) of gene drive applications. But, modification drives do not address overall biting rates since they do not target mosquito density. They might also be vulnerable to the emergence of parasite resistance against the effector mechanism. Combining these two strategies has not been explored sufficiently either by modelling or experimentally. Achieving synergy might not be possible in a straightforward manner due to the opposing nature of their effects on gene flow and genetic population connectivity, especially if the mechanism of sterility or inviability is fully penetrant or the antimalarial effector does not act dominantly.

Gene drives that instead only moderately reduce mosquito fitness allowing for their fixation within populations while also facilitating the propagation of antimalarial effectors could possibly achieve a synergistic combination of population suppression and modification. Such suppression-modification drives must be designed to maintain a desired level of viability and fertility, especially in the hemizygous state, and could simultaneously express an antiparasitic payload. This approach would achieve some degree of population suppression while allowing the drive and also the effector modifications to reach fixation. Indeed, computational modelling supports the notion that modification drives that also reduce fitness can lead to more effective control in some scenarios[19].

To be sustainable, this strategy requires mosquito target genes and gRNA target sites that are supremely conserved yet not essential for mosquito survival or fertility, two seemingly contradictory requirements. A class of genes that are thought to fit these criteria are microRNAs (miRNAs). Many miRNAs encompass highly conserved primary sequences, often maintaining nucleotide sequence identity even across orders, but can evolve to exhibit varying regulatory functions in distantly related organisms[20,21]. Therefore, disrupting miRNAs could affect mosquito fitness to varying degrees while limiting target site resistance. Indeed, systematic studies of microRNA loss of function phenotypes in *Drosophila*, have shown that desirable effects from the perspective of vector control (e.g. a reduction in adult lifespan) are common whereas phenotypes such as total adult sterility or inviability are comparably rare[22].

To validate this approach, we have here chosen to target the *Anopheles gambiae* gene AGAP028779 coding for the conserved microRNA 184 (aga-miR-184). In mosquitoes, miR-184 is among the highest expressed microRNAs often representing a significant fraction of total miRNA or small RNA libraries[23,24]. In *Anopheles* miR-184 is the highest expressed miRNA in the midgut of sugar-fed females and following a blood meal[25]. In addition, mir-184 was also identified as one of the highest-ranking genes in a screen for ultra-conserved genomic regions across 21 mosquito species[26]. Despite its high sequence conservation across metazoans, however, no clear common function has emerged for this gene. Numerous reports from a range of species suggest that miR-184 could play a role in immunity, development or stress responses. In *Drosophila* miR-184 has been implicated in female fertility by modulating female germline stem cell differentiation[27] as well as in other processes related to lifespan and metabolism[28–30] some of which could prove attractive from the perspective of vector control.

In the current study, we demonstrate that a gene drive targeting the non-coding gene mir-184 (miR-184[D]) induces a near perfect inheritance bias in both sexes. We characterise the mir-184 disruption phenotype, uncovering roles for miR-184 in mosquito lifespan, flight, and blood meal metabolism. We show that the miR-184[D] gene drive can effectively propagate itself and a non-autonomous anti-malaria effector in caged populations. Finally, we show through modelling that a combination of reduced lifespan and blood meal-associated mortality can be effective anti-malaria traits.

## Results

### Gene drive construct and transmission efficiency at the mir-184 locus

We generated a gene drive construct where Cas9 was placed under the transcriptional control of the *zpg* germline promoter and expressing a single gRNA overlapping with and targeting a conserved part of mature miR-184 (Fig. 1a). Transgenic strain miR-184[D] was subsequently obtained and we found homozygous male and female individuals to be viable and fertile. We also established the rates of drive inheritance of hemizygous male and female miR-184[D] carriers when crossed to the wild type. We found near-optimal rates of drive transmission in both sexes (Fig. 1b) (♀ 99.6%, and ♂ 97.1%). The inheritance rate from females was significantly higher ($P$ value: 0.002, $N = 1938$), and when evaluated in a subset of crosses we found no significant effect of grandparent drive sex ($P$ value: 0.70, $N = 1170$) consistent with low maternal deposition reported for zpg-Cas9[12,31]. Interestingly, we also observed unexpected GFP expression in the midgut of transgenic adults as well as larvae (Fig. 1c). Dissected sugar and blood fed females (Fig. S1) also revealed substantial levels of GFP expression in the midgut where mir-184 is normally expressed at substantial levels. This modified expression pattern of the neuronal 3xP3-GFP reporter gene suggests that nearby mir-184 cis-regulatory elements were acting to alter transgene expression. We next quantified the levels of the miR-184 miRNA in homozygous, hemizygous and wild-type mosquitoes relative to the levels of the conserved miRNA miR-125. This analysis showed that mature miR-184 is undetectable in whole bodies of miR-184[D]/miR-184[D] females and severely reduced in hemizygous carriers (Fig. 1d).

### miR-184[D] impacts mosquito lifespan and flight ability

We next established a set of measures of mosquito fitness and determined life history traits to evaluate the fitness cost associated with the loss of miR-184 (Fig. 2). In *Drosophila* miR-184 has been reported to regulate germline differentiation and to be necessary for female fertility[26]. However, we detected no significant effect on mosquito fertility and fecundity in various crosses of hemizygous and homozygous miR-184[D] individuals (Fig. 2a). Equally, we found no significant developmental effects in immature stages (Figs. 2b and S2) in contrast to what has been reported in *Drosophila*[32]. Surprisingly we found that a significant number of homozygous adult miR-184[D] individuals (up to 40%) did not spontaneously initiate or sustain flight (Fig. 2b). This phenotype was highly variable and not consistently observed during standard rearing, suggesting that it was the extreme outcome of an underlying distribution of negative fitness associated with the loss of miR-184. We performed flight ability tests on the hemizygous and homozygous miR-184[D] males and females not showing any apparent defect in initiating flight (Fig. 2c). This experiment suggested at least some reduced flight ability amongst homozygous females, but not males. We next analysed mosquito lifespan (Fig. 2d) and found a significant negative effect in sugar-fed homozygous male and female mosquitoes (20 and 25 days median lifespan, respectively) compared

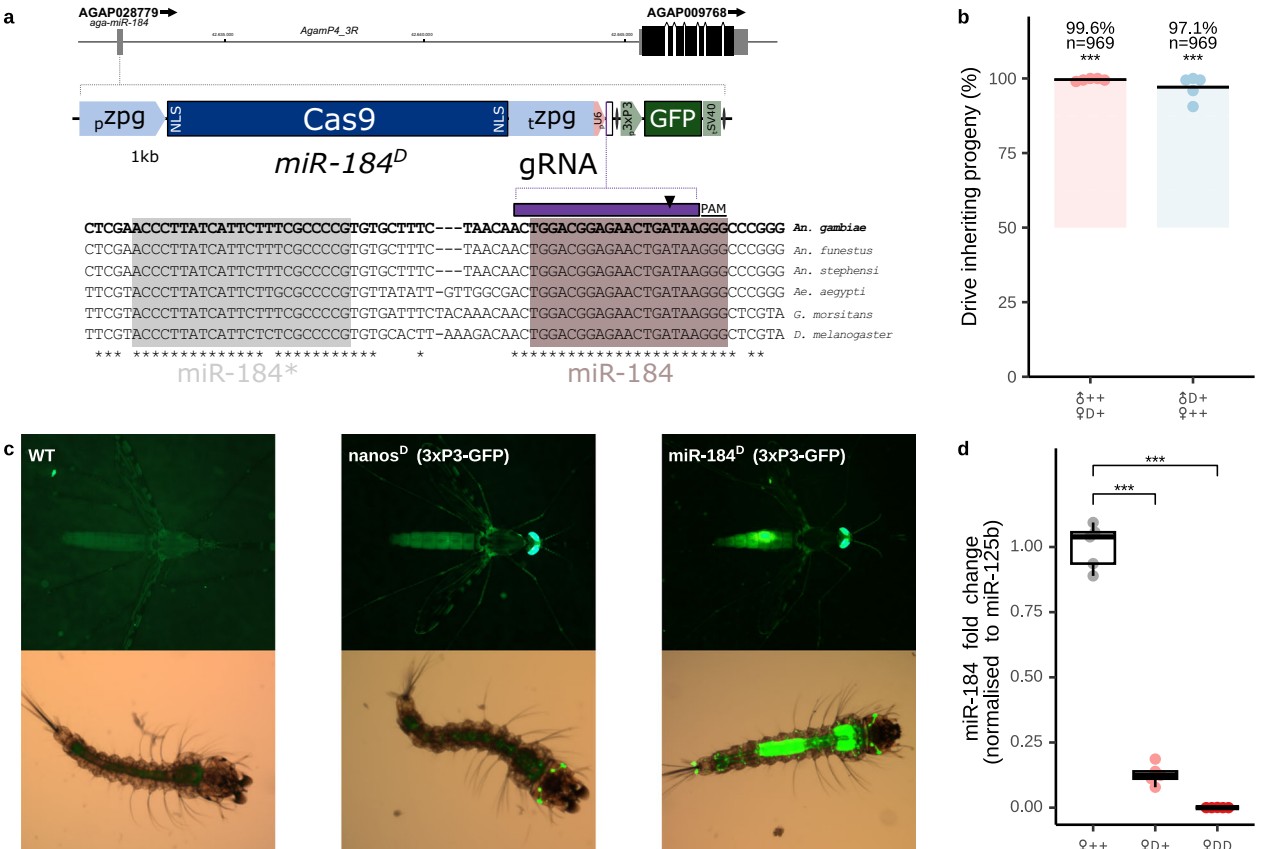

**Fig. 1 | The miR-184^D gene drive. a** Aga-miR-184 (AGAP028779) locus structure, the miR-184^D gene drive construct & gRNA target site conservation across species. **b** Drive transmission in crosses of hemizygous transgenic males or females (D+) to the wild type (++). Each point shows the mean from a pooled independent biological replicate, with the inheritance rates over all replicates and the number of scored progeny indicated. Deviation from Mendelian inheritance rate was calculated using a generalised linear mixed model (*P*\*\*\* < 0.001). Bars originate at the expected inheritance rate and end at the observed overall mean inheritance. **c** GFP expression in transgenic larvae and adult females of the mir-184^D strain and a comparator 3xP3-GFP (nanos^D) transgenic line. **d** Relative miR-184 levels in the whole bodies of sugar-fed females of wild type (++), hemizygous (D+) and homozygous (DD) miR-184^D individuals (*N* = 5 samples) determined by qPCR. Statistical difference from wild type was calculated with Dunnett's multiple comparisons test (*P*\*\*\* < 0.001). Overlaid are box-and-whisker plots of the interquartile range, with a black line indicating the median value. Whiskers extend to the most extreme data point that is no more than 1.5 times the interquartile range.

to the wild-type control (median of 32 and 35 days). A lesser but significant effect on lifespan was also observed in hemizygous individuals.

We hypothesised that the strong ectopic GFP expression in miR-184^D mosquitoes could account for some negative fitness costs we observed. We therefore generated strain miR-184^d that lacks the GFP marker module (Fig. S3). Analysis of the development of homozygous miR-184^d mosquitoes revealed the presence of reduced flight ability as well as a reduced lifespan (Fig. S4), therefore suggesting that these phenotypes are linked to the loss of miR-184 and not the overexpression of GFP. However pupal development was faster in miR-184^d compared to miR-184^D homozygotes indicating that there could be an additional cost of GFP expression also (Fig. S2).

### miR-184^D mosquitos experience blood meal-associated stress

We next performed *Plasmodium* infection experiments with the miR-184^D strain (Fig. S5). Despite its substantial expression in the midgut, our hypothesis was that miR-184^D females would not differ from wild-type females in their ability to support parasite infections. Indeed, we found no significant difference in the oocyst load, oocyst diameter or sporozoite intensity between miR-184^D and wild-type females when infected with *Plasmodium berghei* or *Plasmodium falciparum* (Fig. S5a-e). However, these experiments proved challenging to complete due to the high mortality we observed following the blood meal, occurring with infected as well as uninfected sources of blood (Fig. S5f). We therefore studied the effect of taking a blood meal on the survival of

miR-184^D mosquitoes more systematically. First, different sources of blood including human blood, human blood with heat-inactivated serum as well as cow blood induced significantly higher mortality in miR-184^D females compared to wild-type females (Fig. 3a). We noted that the standard membrane feeding protocol involves the withdrawal of the sugar source from females 3 hours before until 48 h after the provision of a blood meal. This step is intended to guarantee a high rate of blood feeding and the removal of unfed females. We, therefore, sought to determine whether on its own sugar withdrawal or total withdrawal of both sugar and water in the absence of a blood meal would cause increased mortality in miR-184^D females. This experiment (Fig. 3b) revealed no difference between miR-184^D and wild-type mosquitoes. We also carried out a challenge experiment with paraquat, a xenobiotic known to increase the formation of free radicals and oxidative stress[33]. We found that the mortality associated with increasing levels of paraquat exposure was similar in wild-type and miR-184^D females (Fig. 3c). This suggests that oxidative stress induced by the blood meal may not be a primary cause for the observed mortality in miR-184^D females, an otherwise attractive hypothesis given suggestions of a role of miR-184 in the oxidative stress response[34–36]. We repeated our blood-feeding experiments, comparing the survival of blood fed females with or without the withdrawal of sugar (Fig. 3d). In this experiment all females that did not blood feed were manually excluded. We found that when survival was analysed over the entire duration of the experiment there was no statistically significant

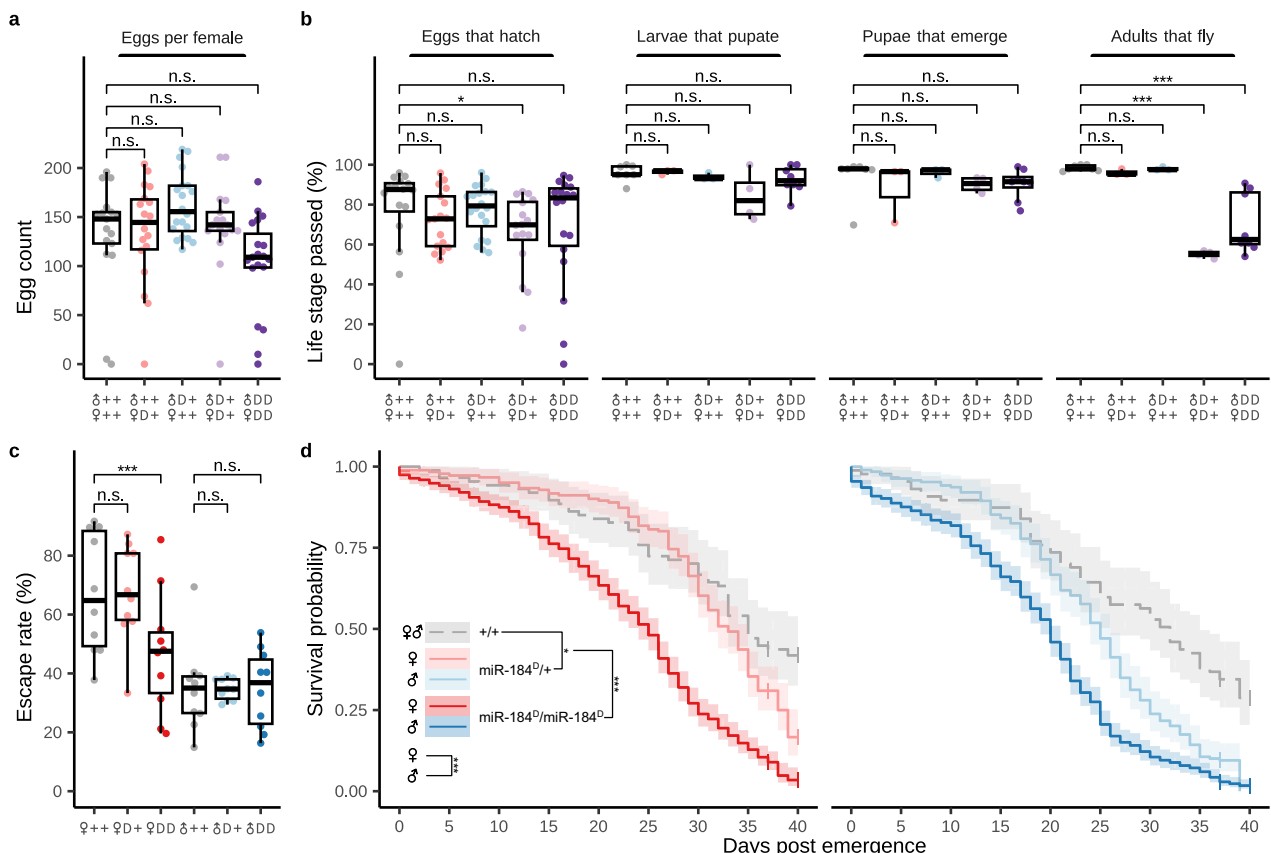

**Fig. 2 | Fitness and life history traits of miR-184^D mosquitoes. a** Fecundity of individual females carrying, or crossed to males carrying, the miR-184^D drive element ($N \geq 17$ individuals). Significance levels were calculated using an unpaired two-tailed Student's $t$-test using Bonferroni correction. **b** Corresponding hatching rates ($N \geq 17$ individuals) and developmental transitions ($N \geq 3$ experiments). DD indicates homozygosity and D+ hemizygosity of the miR-184^D transgene in the parents of the assessed individuals. Significance levels were calculated using a binomial GLM with replicate as a random effect and Dunnett's multiple comparisons test ($P^{ns} \geq 0.05$, $P^* < 0.05$, and $P^{***} < 0.001$). **c** Flight ability using the IAEA/FAO flight test device ($N = 10$ experiments). Significance levels were calculated using a binomial GLM with replicate and time of measurement as random effects and individual

contrasts were performed with multivariate $t$-distribution adjustment of $P$ values ($P^{ns} \geq 0.05$ and $P^{***} < 0.001$). For **a**–**c**, overlaid are box-and-whisker plots of the interquartile range, with a black line indicating the median value. Whiskers extend to the most extreme data point that is no more than 1.5 times the interquartile range. **d** Averaged daily survival of sugar-fed female (left) and male (right) wild-type, miR-184^D homozygous and miR-184^D hemizygous mosquitoes. The miR-184^D groups are composed of multiple separate cross-conditions further broken down in Fig. S4. Survival analysis was conducted using a mixed-effects Cox proportional hazards model. Pairwise comparisons were conducted averaged over the effect of sex using Tukey's method with $P$ value adjustment ($P^* < 0.05$, and $P^{***} < 0.001$). Shaded areas indicate the 95% pointwise confidence intervals.

interaction between the sugar conditions and the miR-184^D genotype ($P$ value: 0.08). However, the immediate mortality (over 8 days, as in the initial experiment in Fig. 3a) following the blood meal was reduced when sugar was always available to miR-184^D females ($P$ value: <0.001). Together these experiments suggest that the blood meal represents a significant stressor for miR-184^D mosquitoes but also that the resulting effect on mosquito survival is likely to be substantially modulated by the nutritional status of females.

**Transcriptional perturbation of solute transport in miR-184^D midguts**

To better understand the diverse phenotypes we observed, we performed transcriptomic analysis on sugar-fed females analysing gene expression levels in dissected midguts and the remainder of the carcass. We identified differentially expressed genes between homozygous miR-184^D and wild-type females and conducted gene ontology (GO) analysis to detect significantly enriched functional categories. (Fig. 3a, b and Supplementary Data 1). Overall, our results indicated a greater extent of transcriptional changes in the gut, in accordance with the expected high level of miR-184 expression in this tissue. We registered only a partial overlap of significantly upregulated (11%) or downregulated (7%) genes between gut and carcass samples (Fig. 3c). GO enrichment analysis

(Fig. 3d) points to solute transport as the most perturbed process in the gut of miR-184^D homozygotes with a range of predicted amino acid, sugar, monocarboxylate, vitamin, water (aquaporin) and ion transporters being significantly upregulated (Supplementary Data 1). Interestingly, when we analysed tissue-specific expression of these genes we found that genes normally enriched for expression in the Malpighian tubules were overrepresented among the significantly upregulated genes in the midgut of miR-184^D females (Fig. 3e). Given that miR-184^D mosquitos have wide-ranging physiological alterations and more work would need to be performed to establish causal relationships between these observations and the loss of miR-184.

We looked for predicted miR-184 targets amongst upregulated genes, cross-referencing our data with two published CLIP-seq datasets[37,38] from *Anopheles* (Fig. S6). This was an attempt to detect a signal from genes directly affected by the loss of mature miR-184 acting upon target mRNA stability. Amongst the predicted 133 high-confidence miR-184 target genes, 20 were found to be among the significantly upregulated genes in the gut samples (Fig. S6a) with septate junction (GO:005918) being the most significantly enriched GO term (adjusted $p < 2.4 \times 10^{-5}$) (Fig. S6b). These included the orthologs of known *Drosophila* septate junction genes undicht (AGAP002289), megatrachea/pickel (AGAP003808), sinuous (AGAP003809) and

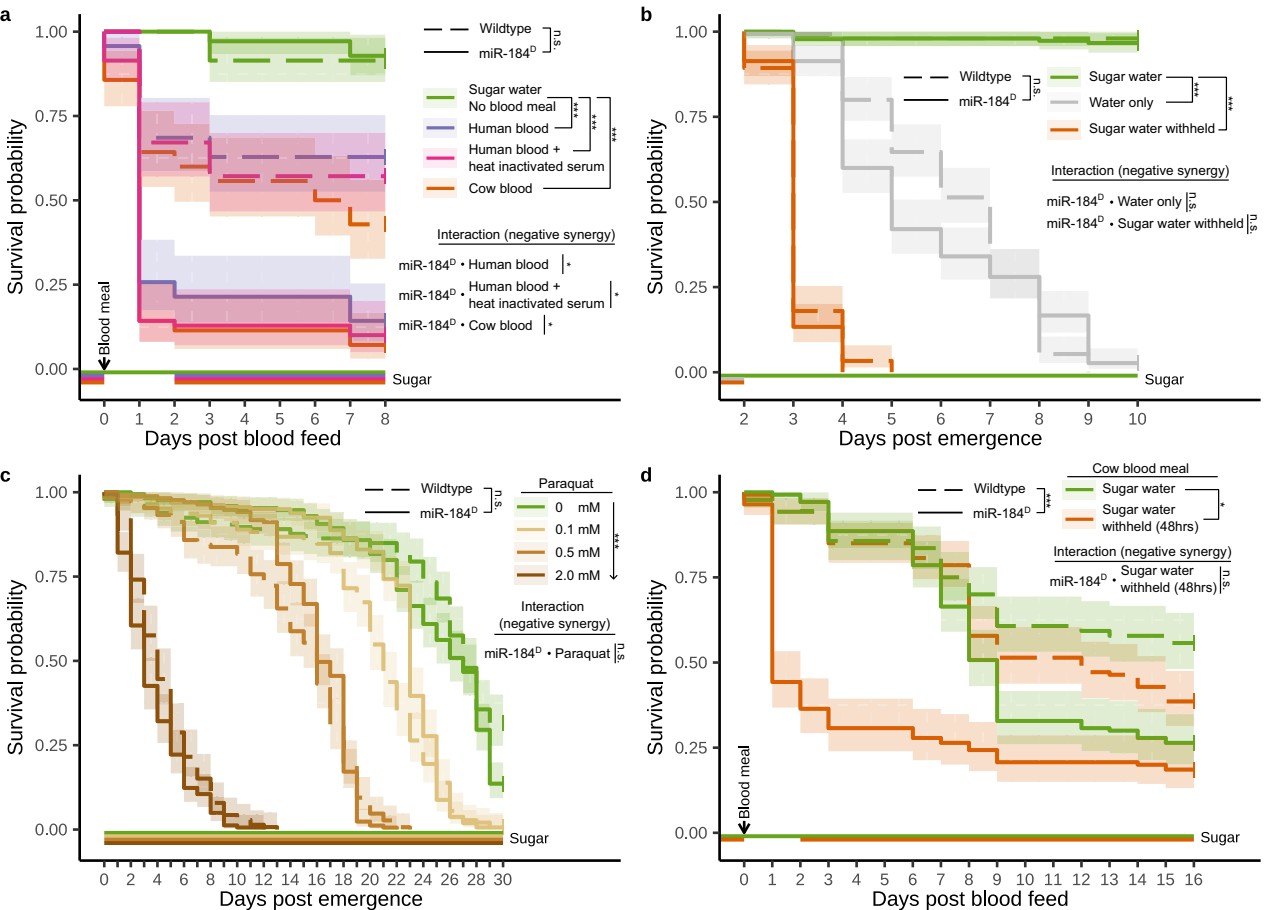

**Fig. 3 | Survival of homozygous miR-184^D and wild-type female mosquitoes following exposure to diverse stressors. a** Averaged daily survival of mosquitoes fed exclusively on sugar versus mosquitoes taking a blood meal associated with two days of sugar withdrawal. **b** Averaged daily survival of mosquitoes under starvation conditions. **c** Averaged daily survival of mosquitoes exposed to oxidative stress by varying levels of paraquat in their sugar water. **d** Averaged daily survival of mosquitoes taking a blood meal with and without the associated withdrawal of sugar. Results include a replicate performed at 21 °C, and temperature was included as a factor in the statistical model. For all panels, survival analyses were conducted using (mixed-effects) Cox proportional hazards models ($P^{ns} \geq 0.05$, $P^* < 0.05$, $P^{**} < 0.01$, and $P^{***} < 0.001$). Sugar water (10% fructose) availability, and when provided, blood meal timing is indicated at the bottom of each panel. Shaded areas indicate the 95% pointwise confidence intervals. The order of each condition in the legend matches the final survival probability ranking of that condition within each genotype.

kune-kune (AGAP003810) the latter three claudin-family proteins having previously been experimentally validated as miR-184 targets[39,40]. All 4 genes also harbour plausible miR-184 binding sites in *A. gambiae* (Fig. S6c).

Amongst the downregulated genes, including thioredoxin peroxidase 4 (TPX4, AGAP011824) a known detoxification factor, we found ornithine decarboxylase (ODC) activity as the only significantly enriched GO term in the gut (Fig. 4d).

Finally, we also used the transcriptomic dataset to look at transgene expression levels (Fig. 4f). In concordance with the results of fluorescent microscopy of transgenic individuals, we found a high level of GFP expression in the miR-184^D gut samples and to a lesser degree in the carcass. However, no analogous modulation of Cas9 expression levels was observed.

Although the phenotype of homozygous miR-184^D mosquitoes was unexpectedly complex, it was in line with our initial requirements for establishing a suppression-modification gene drive. We therefore set out to evaluate miR-184^D in cage invasion experiments next.

## miR-184^D population invasion experiments

The autonomous propagation of miR-184^D and its ability to mobilise an antimalarial payload modification capable of non-autonomous gene drive was studied in six independent cage populations. Unlike classic suppressive gene drives miR-184^D mosquitoes are homozygous-viable and fertile and could be reared in sufficient numbers for large-scale releases. To simulate this, we initiated three cage populations of a total of 500 individuals with homozygous transgenics (50♂, 50♀) and wild type mosquitoes (200♂, 200♀) to achieve a miR-184^D starting allele frequency of 20%. Three further cage populations contained in addition to miR-184^D the MM-CP transgene also at a 20% starting allele frequency. The MM-CP transgene (located 5.87 Mb upstream of the mir-184 locus on chromosome 3) first described in Hoermann et al. expresses two antimicrobial peptides and has been shown to retard *Plasmodium* sporogonic development[15]. It also expresses a gRNA, but is only able to bias its own inheritance in the presence of Cas9, which here is provided in trans by the miR-184^D drive.

We monitored the frequency of both transgenes over time and also recorded the cage egg output for each generation. Although egg output is an imperfect measure of the varied fitness effects of miR-184^D it was chosen for simplicity and should in compound also reflect factors such as increased mortality or reduced flight ability. We observed an initial reduction in the fraction of miR-184^D transgene carriers in 5 out of 6 cage populations (Figs. 5a, b and S7) in line with lower homozygous fitness of miR-184^D individuals presumably resulting in a

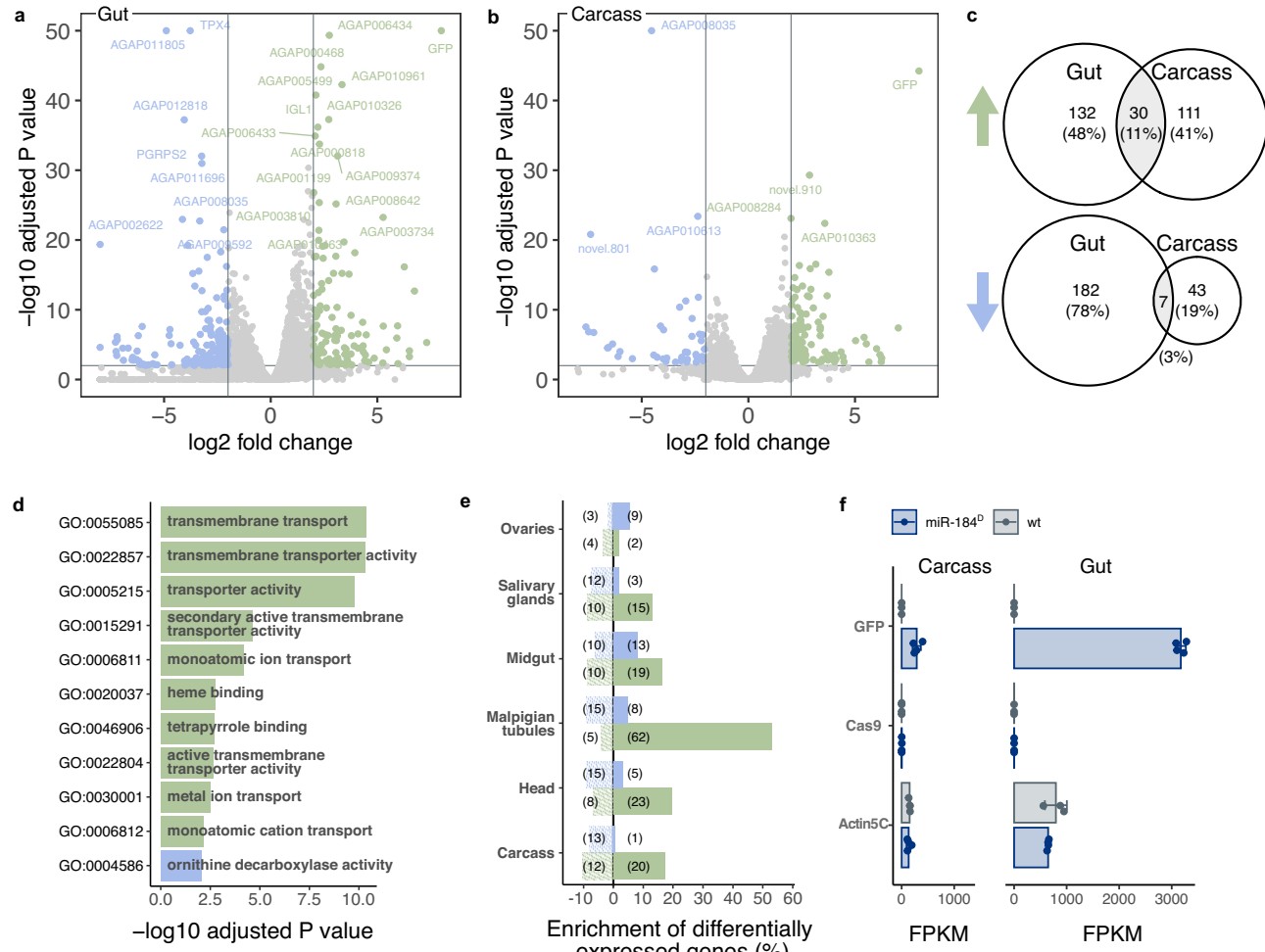

**Fig. 4 | Transcriptomic analysis of homozygous miR-184$^D$ and wild type female mosquitoes.** Volcano plots of RNAseq experiments performed on midguts **a** and the rest of the body **b** of sugar-fed females. Differentially expressed genes between miR-184$^D$ and wild type mosquitoes ($P \leq 0.01$ and log2-fold change $\geq 2$) are indicated. *P* values are Benjamini-Hochberg adjusted. **c** Venn diagrams showing the total number of differentially expressed genes and overlap between the two tissues. **d** Summary of enriched GO terms associated with significantly upregulated and downregulated genes in the gut samples. *P* values are Benjamini-Hochberg adjusted. **e** Tissue-enrichment calls of significantly upregulated and downregulated genes in the gut samples using the Baker et al. dataset[58]. **f** Analysis of transgene transcript levels ($N \geq 3$ samples), presented as mean FPKM values ± SD.

reduced participation in mating and a reduced egg output of generation 0. Consequently, generation 1 featured the lowest point in the gene drive allele frequency and maximum heterozygosity observed and also the highest egg output on average (Fig. 5c, d). Subsequently, both transgenes increased in all caged populations and reached fixation around generation 14. We reassessed the composition of all 6 cages in generations 20 and 21, and in generation 21 we also performed a detailed analysis by DNA sequencing of all miR-184$^D$ and CP alleles whose size did not match the drive allele (Fig. 5e). No wild-type alleles for both loci were detected in any of the 6 cage populations, and only 1 out of 552 individuals screened did not carry at least one allele of miR-184$^D$. For miR-184$^D$ two cage populations featured non-drive alleles, the most prevalent (detected in 20 individuals in cage population 4) being a 5 base pair deletion mutant. However, the nature of miRNA function limits inferring function from sequence data in a straightforward manner. For the CP locus, the target gene into which the MM-CP transgene homes, which was genotyped using PCR throughout the entire population cage experiment (Fig. S8), we detected low-frequency non-drive alleles in cage populations 5 and 6. DNA sequencing revealed that the most prevalent variant allele, a triplication of a GATT motif, is likely a functional pre-existing rare variant of the CP 5' UTR. Overall, egg output fluctuated substantially across

generations with all 6 cage populations generally affected the same way (Fig. 5c, d). This suggests that, during this year-long experiment, the blood source or other environmental parameters we could not control for had an outsized influence on population productivity. Nevertheless, the propagation of the miR-184$^D$ drive and its eventual fixation lead to a corresponding steady decrease in egg production over time. Finally, our data indicates that miR-184$^D$ was highly effective in propagating the antimalarial effector MM-CP in all 3 cage populations.

## Gene drive and malaria transmission model

To evaluate the impact of a gene drive intervention that would reproduce the miR-184$^D$ drive phenotypes that we observed in the lab on malaria transmission in the field we performed simulations using the EMOD framework[19,41]. This agent-based framework incorporates entomological and epidemiological dynamics to evaluate a range of anti-malaria interventions including gene drives. To account for varying degrees of possible resistance and phenotype levels displayed by mir-184$^D$ in a field setting we evaluated a range of parameter combinations (Fig. 6a). Assuming 99% inheritance of the drive we varied the ratio of the generation of functional (R1, values evaluated: .1%, .01%, .001%) to non-functional resistance alleles (R2). The potentially diverse fitness

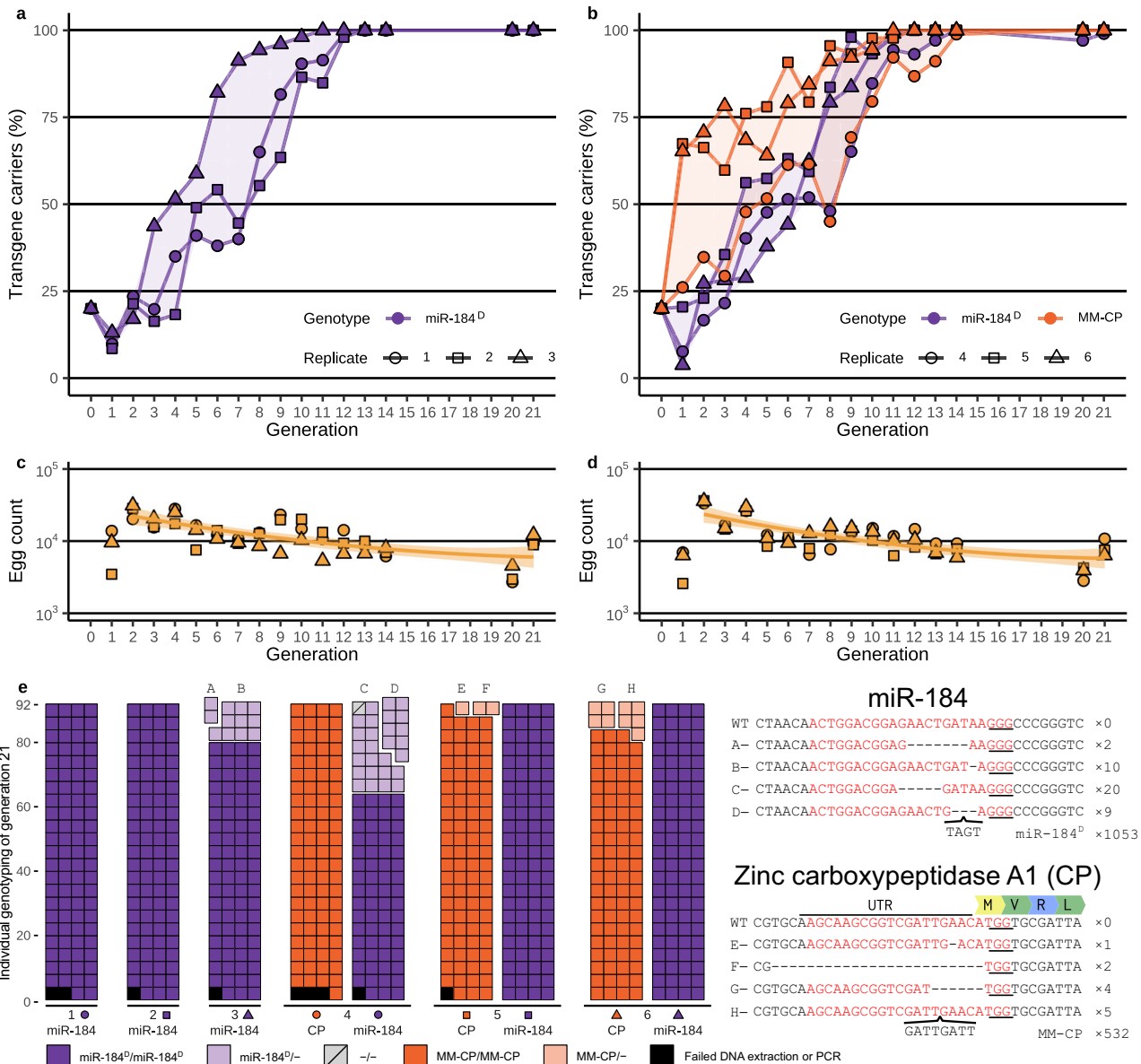

**Fig. 5 | Population invasion experiments.** 3 cage populations (1–3) seeded with the miR-184[D] transgene at a 20% starting allele frequency and 3 cage populations (4–6) seeded with the miR-184[D] and the MM-CP transgenes at 20% starting frequencies. Shown in **a** and **b** is the transgene carrier frequency over multiple generations determined by fluorescent (GFP miR-184[D]) and molecular genotyping (markerless MM-CP). Panels **c** and **d** indicate the egg output per generation of the respective cage populations. A quadratic trendline was fitted to the combined egg output of the three replicates per cage condition over the generations, excluding the setup generation in which all transgene carriers were homozygous. The shaded region shows the standard error (SE) of the fitted line. **e** Composition of alleles in generation 21 at the miR-184 and CP loci in all 6 populations determined by PCR and subsequent DNA sequencing of non-drive alleles.

impact of miR-184 loss of function (LOF) were modelled as an increase of adult daily mortality in homozygotes (ranging from 1–2.5×) and heterozygotes (50% lower impact: 1–1.75×) relative to functional miR-184 homozygotes. We further modelled the full range of blood meal induced mortality (0–100%) in mosquitoes with a homozygous miR-184 loss of function genotype. A single release of 1000 male mosquitoes homozygous for miR-184[D] was performed after which the impact was evaluated over 5 years sweeping over three different vector density scenarios (-15, 35, or 65 infectious bites per person per year) averaged across 40 stochastic simulations. We predicted the reduction in clinical malaria cases (Fig. 6b) as well as the probability of eliminating the malaria burden locally (Fig. S9) under the assumption of *A. gambiae* being the sole malaria vector present.

High levels of increased daily mortality interfere with reproduction and thus impair the propagation of the drive (Fig. S10); conversely we observe fixation and plateauing of the drive, analogous to what we observed in our cage trials, when the probability of functional resistance was low and the combined effect of daily and blood meal induced mortality was moderate or low (Fig. S11). Mosquito population eradication only occurred in our simulations when blood meal mortality was 100% and functional resistance was low (0.001%, 207 simulations) or medium (0.01%, 3 simulations) (Fig. S12). At mortality levels where the drive can spread effectively, the increased daily mortality rate and even more so the sex-specific blood meal associated mortality reduces the proportion of female mosquitoes that live long enough to transmit the parasite (Fig. S13). Notably, intermediate blood meal and daily mortality

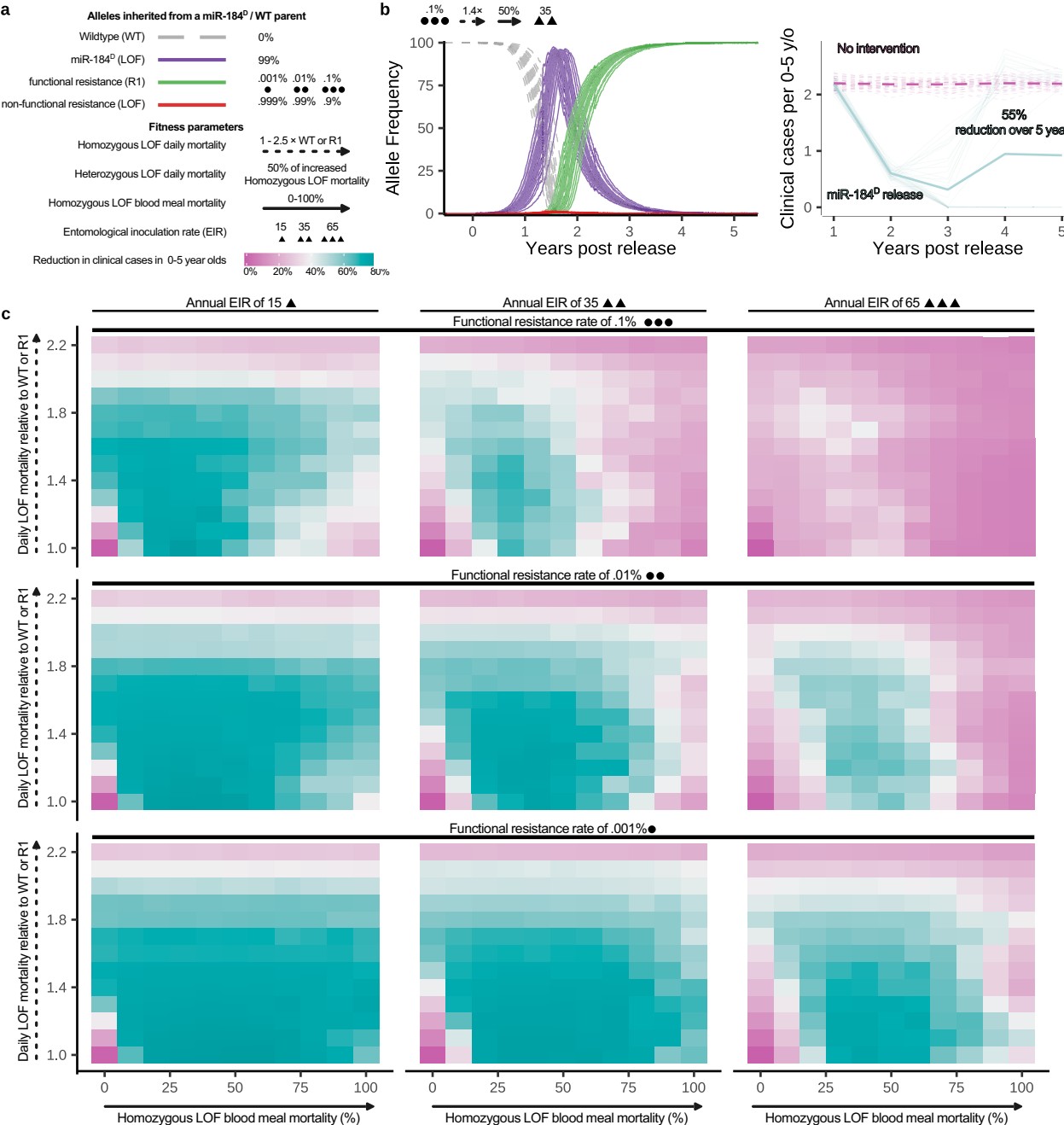

**Fig. 6 | Modelling gene drive propagation and its effect on malaria transmission. a** Gene drive and fitness parameters. The four parameters varied for each set of stochastic simulations are indicated. Those are, the rate of functional resistance rate (circles), loss of function daily mortality rate (dashed arrow), loss of function blood meal mortality rate in homozygotes (solid arrow) and the entomological inoculation rate (triangle). **b** An exemplary miR-184[D] allele frequency and clinical case dynamics plot for one set of parameter combinations. Each parameter combination was run 40 times. **c** Overall malaria reduction was measured over 5 years following gene drive releases compared to the no-release controls. Outcomes are averaged over 40 stochastic simulations.

rates more specifically impact the proportion of long-lived female mosquitos, compared to a more general population suppression at high mortality rates. Under the high functional resistance scenario this more targeted fitness cost reduces the selection pressure for functional resistance and results in an overall higher impact on malaria cases. The malaria burden is heavily dependent on the lifespan of female mosquitoes as transmission requires at least two blood meals separated by the *Plasmodium* incubation period. For this reason, the fitness effects imposed by the loss of miR-184 when modelled across a range of parameters on the population level are predicted to frequently be able to eradicate the local malaria burden (Figs. 6b and S9) and result in

strong reductions in clinical cases averaged over the 5 years following releases (Fig. 6c).

## Discussion

No gene drive has been tested in the environment; however, for *Anopheles* mosquitoes, gene drive technologies have reached a stage where they can rapidly propagate through laboratory populations resulting in the elimination of that mosquito population or spreading of traits that target the parasite[12,14,16,17]. Efforts are underway to develop suitable technical, social, and regulatory processes for field testing and to facilitate potential deployment in the coming years[42-46]. A range of

designs for population modification and population suppression drives have been suggested and described in the literature. In the present study, we sought to explore the possibility of designing robust drive strategies that would combine aspects of these two strategies i.e. the reduction of a mosquito population's fitness and the simultaneous reduction of the individual vectorial capacity of mosquitoes.

Our choice of the target gene was driven by the supreme sequence conservation of miR-184 and its high expression levels in the midgut, a favourable tissue for exposing parasites to the products of coexpressed effectors in mosquito vector species. Our experiments revealed no evidence of a direct role of miR-184 in mosquito female fertility, larval development or in orchestrating the response to oxidative stress as had been previously suggested by a number of studies in other organisms including *Drosophila*. Instead, we find that the loss of miR-184$^D$ triggers a complex set of negative fitness effects such as a reduced adult lifespan and flight ability or the disability to deal with the stress associated with a blood meal. It has been shown that miRNAs expressed at higher levels and in more tissues and cell types regulate a larger set of target genes leading to greater functional constraints on miRNA sequence due to pleiotropic effects of deleterious mutations[47]. This would link the pleiotropic phenotypic effects, broad expression at high levels to the strong sequence conservation of miR-184.

Our subsequent investigation into the biology of miR-184 in *Anopheles* supports the view that it plays a role in regulating permeability barriers, possibly modulating transcellular and paracellular solute transport. As such miR-184 could also contribute to maintaining tissue identity in the digestive and excretory organs of the mosquito. We found that the mosquito midgut of homozygous miR-184$^D$ transgenics over-expresses a number of genes normally associated with the Malpigian tubules. For adult female mosquitoes, the blood meal is a particularly dramatic metabolic event that causes the ingestion of more than the equivalent of their body mass in blood. Our overall hypothesis is therefore that, due to the disruption of these transport processes, mosquitoes lacking miR-184 are unable to deal with the metabolic stress of a blood meal or, alternatively, that they fail to utilise the blood meal fully. Our data hints that miR-184 could act in the first instance via regulating the expression of septate junction components, some of which we identified as potential miR-184 target genes in *Anopheles*. This aligns with findings in *Drosophila* where miR-184 had been implicated in the regulation of septate junction genes contactin, coracle, kune-kune, megatrachea/pickel, nrxIV, sinuous and as well as gliotactin[39,40]. To fully elucidate the function of miR-184 and its relationship to the phenotypes we observed in mosquitoes, further and more detailed studies are necessary. The ΔmiR-184 gene drive strain we have generated will be an invaluable tool towards this end. Although the function of some miRNAs has been studied in *Aedes* mosquitoes using functional genetics[48,49], the miR-184$^D$ strain represents the first knock-out strain of a miRNA gene in malaria vectors.

The phenotypic effects in miR-184$^D$, largely manifesting themselves in homozygotes, were predicted to reduce vector fitness and some effects were attractive from the perspective of vector control, for example, the reduction in adult lifespan which would directly affect vectorial capacity. In addition, we found no effect on mosquito fertility and high rates of drive. We therefore performed population cage experiments and showed that the miR-184$^D$ construct could rapidly invade caged *A. gambiae* populations and that its fixation resulted in correspondingly reduced population fitness. Such a dynamic, i.e. sustained fixation of a costly drive had not been previously described.

It is challenging to predict how the complex fitness profile of miR-184$^D$ mosquitoes might translate to field conditions. This could be a disadvantage of choosing a regulatory molecule such as a miRNA as a gene drive target, compared to genes causing more straightforward phenotypes such as the loss of female fertility or the disruption of sex determination. To begin addressing this using the EMOD modelling framework, we simulated general fitness impacts of miR-184 loss of function (daily mortality and blood meal mortality) and evaluated a wide range of possible parameter values. The miR-184$^D$ blood meal associated mortality we observed experimentally is severe, and attempts have been made by others to directly engineer it as a suppression trait[50]. However, we show that for miR-184$^D$ it is contingent on the nutritional state of the mosquitoes, likely lowering its penetrance. Nonetheless, our modelling indicates that even low levels of blood meal mortality in combination with overall reduced vector lifespan are effective at lowering malaria transmission, and is more effective than full lethality when resistance selection is factored in. Moreover, we found that, at least under laboratory conditions, miR-184$^D$/miR-184$^D$ homozygote mosquitoes are viable and fertile and miR-184$^D$ can thus be reared as a homozygous true breeding strain, an advantage over canonical suppression strains that need to be continuously back-crossed to the wild-type which in turn complicates the generation of target product profiles and quality control steps.

We observed cleavage-resistant miR-184 alleles in 2 of the 6 cage populations. To what extent these miR-184 allele variants also represent functional or partially functional miRNA alleles we did not address here given the identification of only a single non-transgenic mosquito and the complexity of the ΔmiR-184 phenotype. It is possible that the mutant alleles we identified in multiple individuals retain some miR-184 function, however, population dynamics after the exhaustion of all wild-type target alleles could also feature other selection events. For example, the selection between the fitness of drive alleles that actively express GFP and Cas9, and indel alleles that do not, both of which would be Δmir-184 alleles. This is particularly relevant as in our experiments some cage populations were studied for up to 10 generations beyond the initial fixation of the drive where such a dynamic could unfold. In any case, a field-ready miR-184 gene drive would likely feature a secondary or tertiary gRNA[13,14,17,51]. Ideally, these would also direct cleavage to the seed region of the microRNA for improved robustness, which is not the case in our pilot design.

While we observed and studied some phenotypes affecting the fitness of miR-184$^D$ transgenics in the laboratory, it is certain that other such fitness effects exist, some of which would only manifest themselves fully or at all under real-life conditions in a varied environment. Follow-up studies would thus be needed to further dissect the complex fitness effects and to evaluate miR-184$^D$ mosquito fitness under semi-field conditions e.g. for the purpose of compiling a risk assessment. It will also be key to determine how effective our strategy of targeting an RNA gene is in reducing the likelihood of functional resistance emerging. If larger-scale experiments indicate that functional resistance occurs at low rates, then our modelling suggests a wide range of fitness effects will lower malaria transmission. Inclusion of an anti-malaria effector may also address the unlikely case that the impact of miR-184 disruption is less severe in field settings than in laboratory conditions, and increase the impact of the drive in high transmission settings.

The miR-184$^D$ drive we describe combines aspects of population suppression and population modification. An example of a suppressive effect it exerts is the mortality following the blood meal leading to lower reproductive output. However, the drive persisted at high allele frequencies in caged populations which, together with its effect on mosquito lifespan, which directly impacts vectorial capacity in the field is more akin to a modification drive. Going beyond this, we also used miR-184$^D$ to co-propagate a non-autonomous separate antimalarial effector, thus achieving a further combination of suppression and modification in these caged populations. Another alternative is presented by the high expression levels of the miR-184 microRNA gene in the *A. gambiae* midgut. We observed strong GFP expression in the gut of transgenics even though the marker gene was under the control of the neuronal 3xP3 promoter. This points to the action of powerful cis-regulatory elements at the miR-184 locus and suggests that this locus would lend itself well for the direct and robust expression of anti-malarial effectors in the mosquito e.g. single-chain antibodies

targeting *Plasmodium* surface factors[5,16]. With the addition of a suitable effector module, this drive would then constitute a single-locus suppression-modification construct.

In conclusion, we describe a miRNA targeting gene drive that has attractive attributes and can integrate aspects of both population suppression and population modification. This candidate strain should be evaluated further as a tool for malaria control.

## Methods

### Design and generation of constructs

An annotated DNA sequence file for the final transformation construct pD-miR184D is provided (OSF repository). Briefly, a gRNA target site was identified with CRISPOR[52] and high conservation was confirmed with the Ag1000G database[53]. The ~800 bp up and downstream from the gRNA cut-site were PCR amplified from a wild-type Ifakara mosquito colony[54]. The zpg promoter, Cas9 CDS and zpg terminator were PCR amplified from 105-pD-zpg-Cre-Cas9 (Hoermann et al., unpublished), and the *A. gambiae* U6 promoter (AGAP013695), and 3xP3-eGFP-SV40 flanked by LoxP sites were PCR amplified from the intermediate plasmid 106-pI-3UTR-gRNA-GFP (Hoermann et al. unpublished). Some fragments were pre-fused via overlap-extension PCR and the final plasmid was generated via Gibson assembly. *A. gambiae* Ifakara eggs were injected[55] with the donor plasmid and the p155 helper plasmid[11], giving rise to 31 transient G0 larvae. Of these, 19 were male and 9 female, and the rest did not survive to adulthood. The 9 females were crossed to wildtype and gave rise to 13 GFP-positive progeny and subsequently to 23 G1 transgenics. Correct integration was confirmed by Chelex gDNA extraction and PCR over the 5′ and 3′ junctions with primer pairs 665 + DE365 and 449 + 666 and subsequent Sanger sequencing. The miR-184[d] line was established by crossing homozygous miR-184[D] individuals to a DsRed marked zpg-Cre-Cas9 line (Hoermann et al. unpublished) within the Ifakara background. This resulted in the excision of the loxP-flanked GFP marker. The trans hemizygous progeny were intercrossed, and their non-GFP & non-DsRed offspring were selected and again intercrossed. Genotyping of the pupal cases was used to establish homozygous strain miR-184[d].

### Mosquito husbandry

All mosquitoes were maintained at ~27 °C and ~70% humidity with a 12 h:12 h light:dark cycle and provided with 10% Fructose *ad libitum*. All sexing was performed at the pupal stage. Mosquito aquatic stages were kept in $25^2$ cm trays with ~500 ml of water and adults were housed in $17.5^3$ cm cages, or $25^3$ cm in the case of the cage trial. The Ifakara colony was used as wild-type control for all experiments. All experiments were performed with cow blood (First Link (UK) Ltd.), unless otherwise stated.

### Microscopy and dissections

Homozygous female miR-184[D] mosquitoes were dissected 8 days post blood meal. The digestive and reproductive systems were isolated and imaged by transmission and fluorescent microscopy.

### miRNA quantification

Total RNA was extracted from individual mosquitoes homogenised in 250 μl Trizol (Thermofisher) following the manufacturer's instructions. For each individual, 30 ng RNA was reverse transcribed using a miR-CURY LNA RT Kit (Qiagen). qPCR was performed using miRCURY LNA PCR assays for aga-miR-184-3p (YP02115202) and hsa-miR-125b-5p (YP00205713) and a miRCURY LNA SYBR Green PCR Kit (Qiagen) from 50X diluted cDNA as per manufacturer's instructions in a 7500 Fast Real-Time system (Thermofisher).

### Assessment of autonomous gene drive

For each replicate ($N = 5$) hemizygous miR-184[D] males or females were crossed to wild types (pool sizes of 20–76) and 200 of their progeny were collected and subsequently screened for the presence of the 3xP3-

GFP marker. Estimated means and 95% confidence intervals were calculated by a generalised linear mixed model, with a binomial ('logit' link) error distribution fitted using the glmmTMB package[56]. Results were averaged over the levels of grandparent drive carrying sex (Female, Male, Mixed), with replicate as a random effect. Statistical significance was calculated using Tukey HSD. When comparing grandparent sex results were averaged over the levels of parent sex (Male, Female). For assessing inheritance bias in the markerless line, confirmed homozygous miR-184[d] male mosquitoes were crossed to wild-type females. Their hemizygous male or female offspring were crossed to wild types (pools of 50 males and females, two replicates) and in each replicate 184 of their progeny were collected and individually genotyped for inheritance of the miR-184[d] transgene and the presence of a wild-type allele (as a control for the PCR and DNA extraction) using primers DE375, 680, 729. PCRs that failed to give products were repeated with another set of primers (1 out of 736 failed both attempts).

### Fitness assays

For the fecundity and fertility assays, cages containing 50 females and 50 males were let to mate for 5 days and blood fed with cow blood. A day after the blood meal, single blood-fed females were transferred to cups containing water, lined with filter paper. The second day after the blood meal, dead females in the cups were substituted. Females that failed to lay eggs or produce larvae after the second day were dissected and excluded from the analysis if no sperm was detected in their spermatheca. Egg counts were compared to the wild type using *t*-tests with Bonferroni correction for multiple testing. Hatching rate significance levels were calculated using a binomial GLM with replicates as random effects and compared by Tukey's HSD. Separately, larvae for the different genotypes were counted and scored every day through different life stages. Adults were considered non-flying if they did not readily leave the water surface or started but did not complete the process of emerging.

### Determination of mosquito flight ability

Flying male and female adult mosquitoes were assessed for their flight ability with a 22 × 10mm IAEA/FAO flight test device[57] (Cicindela Ltd, Valencia, Spain). Tests were performed on pools of about 50 individuals of either sex with $N = 10$ replicates for each genotype and sex, in either the morning or afternoon. Pools were placed in a tight cup, and chilled for 45 seconds on ice, before being placed at the bottom of several vertical tubes topped by a fan and BG-Lure Mosquito Attractant (Biogents). The escape rate was calculated as the number of individuals that left the bottom chamber at the end of two hours. Significance levels were calculated using a binomial GLM, averaged over the time of day with replicate as a random effect.

### Pupation timing and adult survival assays

Various crosses were performed to synchronise the hatching of wild type, two types of hemizygous and three types of homozygous mosquitoes for assaying pupation timing and adult survival: 1: Individuals of the wild type colony were crossed together. 2: Homozygous miR-184[D] males were crossed to wild-type females to generate hemizygous individuals. 3: Wild-type males were crossed to homozygous miR-184[D] females to generate hemizygous individuals. 4: Homozygous miR-184[D] individuals were intercrossed to generate homozygous offspring. 5: Hemizygous miR-184[D] males were crossed to hemizygous miR-184[D] females to generate largely homozygous offspring (due to the very high inheritance rates of miR-184[D]). 6: Markerless homozygous miR-184[d] individuals were intercrossed to generate homozygous offspring. On the day of hatching, larvae were distributed over 3–6 trays of ~100 larvae each (two replicates), and checked for pupae each day, which were then removed and sexed. Dead larvae were excluded. Pupae of each of the crosses and sex were allowed to emerge into cages aiming for 50 adults capable of flying in each cage. The mosquitos were provided with 10% fructose

solution which was periodically refreshed when at low levels throughout the experiment. Dead mosquitoes were counted daily and removed.

## Mosquito infection assays

Homozygous miR-184$^D$ or wild-type mosquitos were infected with mature *P. falciparum* NF54 gametocyte cultures (2 to 6% gametocytemia) using a streamlined standard membrane feeding assay or with *P. berghei* ANKA 2.34 by direct feeding on infected mice. Engorged mosquitoes were provided 10% sucrose and maintained at 27 or 21 °C for *P. falciparum* and *P. berghei* infections, respectively. For infections with *P. falciparum*, mosquitoes were starved for 48 hours after the infective or supplemental blood meal to eliminate unfed individuals.

## Mosquito infection survival assay

Homozygous miR-184$^D$ mosquitos were sexed and left to emerge in a cage with 10% fructose solution. Adults were transferred to cups and provided sugar solution or uninfected, heat-inactivated (HI), or a *P. falciparum*-infected blood meal with a two-day sugar withdrawal. Sugar was withdrawn ~1 hour before blood feeding. Mosquitos that were not engorged after two days were removed and excluded from the survival analysis. Significance levels were calculated using a binomial GLM.

## Blood type feeding survival assays

Homozygous miR-184$^D$ mosquitos were sexed and left to emerge in a cage with 10% fructose solution. Adults were transferred to cups and provided a sugar solution or a two-day sugar withdrawal with a blood meal of human blood, human blood with heat-inactivated serum, or cow blood. Sugar was withdrawn ~1 h before blood feeding. Mosquitos that were not engorged after two days were removed and excluded from the survival analysis. Accumulating dead mosquitoes were counted daily, apart from days 4 and 5. Significance levels were calculated using a Cox proportional hazard model.

## Sugar-feeding survival assays

For each strain and condition, pupae were sexed and females were left to emerge in cages with a supply of 10% fructose. In each cage, adults in excess of 50 were removed and the next day the sugar source was removed, replaced with demineralised water (dm), or sugar was left in place for the control condition. Accumulating dead mosquitoes were counted daily. Significance levels were calculated using a mixed-effects Cox proportional hazard model.

## Paraquat feeding survival assays

For each strain and condition, pupae were sexed and females were left to emerge in cages with a supply of 10% fructose. In different cages, the sugar solution contained increasing concentrations of Paraquat (N,N'-dimethyl-4,4'-bipyridinium dichloride, Sigma) (0, 0.1, 0.5, 2.0 mM). Accumulating dead mosquitoes were counted daily. Significance levels were calculated using a mixed-effects Cox proportional hazard model.

## Blood feeding with sugar survival assays

Homozygous miR-184$^D$ mosquitos were sexed and left to emerge in a cage with 10% fructose solution. Adults were transferred to cups and provided a cow blood meal, with or without a two-day sugar withdrawal. Sugar was withdrawn ~1 hour before blood feeding. Mosquitos in the blood meal conditions that were not engorged after two days were removed and excluded from the survival analysis. Survival was assessed over 16 days at 21 and 27 °C. Accumulating dead mosquitoes were counted daily, apart from days 4, 5, 10, and 11. Significance levels were calculated using a Cox proportional hazard model.

## RNAseq analysis

Sugar-fed females were dissected into Trizol. After homogenisation with 2.8 mm ceramic beads (CK28R, Precellys), RNA was extracted with the Direct-zol RNA Mini-prep kit (Zymo Research) including on-column DNase treatment. Four biological replicates per condition were subjected to RNA-seq. Libraries were prepared with the NEB Next® Ultra™ RNA Library Prep Kit and sequenced on a NovaSeq 6000 Illumina platform (instrument HWI-ST1276) generating 150 bp paired-end reads. Sequencing reads were mapped to the *A. gambiae* PEST genome (AgamP4.13, GCA_000005575.2 supplemented with the miR-184$^D$ construct reference) using HISAT2 software v2.0.5 (54) (with parameters --dta --phred33). Differential expression was assessed with DESeq2 v1.20.0. and GO enrichment analysis was performed using g:Profiler using a *P* value cutoff of *P* = 0.01.

## Population cage setup and maintenance

Age-matched male and female pupae were allowed to emerge in separate cages and mixed as adults. For cages 1-3, this was 200 wild types and 50 miR-184$^D$ homozygotes for each sex. For cages 4-6 this was 150 wild types, 50 miR-184$^D$ homozygotes, and 50 markerless MM-CP homozygotes (described in Hoermann et al.[14]) for each sex. For each generation of the cage trials, mating proceeded for ~6 days, adults were blood fed and two days after, the mosquitoes were allowed to oviposit into two egg bowls filled with water and lined with filter paper and eggs were bleached for 1 minute with 2% bleach. After egg collection and hatching, for each cage, 5 trays with 105 larvae were maintained to seed the next generation of the cage trial.

## Population cage screening

For generations 1–14, 20, and 21 of the cage trials, eggs were photographed and counted using MECVision with the following parameters: Image Threshold Adjustment: 30; Minimum Egg Size: 1; Maximum Egg Size: 2; Maximum Cluster Size: 15. Subsequently ~100 larvae separate from those used to seed the next cage generation were screened under a fluorescent microscope for the presence of the miR-184$^D$ GFP marker. For cages 4-6 that include the markerless MM-CP transgene, after being screened for GFP, larvae were distributed over a 96-well plate (92 larvae and 4 controls) and gDNA was extracted with the Phire Tissue Direct PCR Kit (Thermo Scientific). Multiplex PCR was performed with primers 260, 531, and 532, and were genotyped by size discrimination on a gel. For generations 1 and 20, the gDNA plates were also used to genotyped the miR-184 locus using primer 729, 735, and 680. For genotyping generation 21, primers 1210, 735, and 1216, were used which avoided sequence variation present in our wild type lab colony (which was confirmed to not have impacted the outcome of previous genotyping assays).

## Statistics, visualisation, and reproducibility

Data analysis and visualisation were performed with R (R Development Core Team), apart from Fig. S5a–f for which GraphPad Prism was used. Code to reproduce plotting and statistical analysis performed in R is provided. Gene diagrams and survival curve legends were generated with Inkscape.

## Statistics and reproducibility

No statistical method was used to predetermine sample size. No data were excluded from the analyses. The experiments were not randomised, and investigators were not blinded to allocation during experiments and outcome assessment.

## Gene drive and malaria transmission modelling using EMOD

Simulations were performed with EMOD v2.23, a mechanistic, agent-based model of *P. falciparum* malaria transmission that includes vector life cycle dynamics and within host-parasite and immune dynamics[19,41]. Seasonality of rainfall and temperature as well as vector species were kept the same across transmission settings, but vector density was varied to match desired transmission intensity. Each simulation contained approximately 1000 representative people with birth and death rates appropriate to the demography without considering importation of

malaria. We include baseline health seeking for symptomatic cases as an intervention where human agents can seek treatment with 80% artemether-lumefantrine of the time within 2 days of severe symptom onset and 50% of the time within 3 days of the onset of a clinical but nonsevere case. Mosquitoes within EMOD contain simulated genomes that can model up to 10 genes with eight alleles per gene with phenotypic traits that map onto different genotypes[41]. The gene drive modelled here is composed of four separate alleles: WT, miR-184$^D$, functional resistance, and non-functional resistance. The miR-184 and non-functional resistance alleles, and WT and functional resistance alleles were identical in regards to their fitness cost. The two sets of alleles only differed in the special case where a miR-184$^D$ allele was paired with a WT allele, allowing inheritance bias to occur (99% miR-184$^D$ inheritance) and new resistance alleles to be generated from the remaining 1% of alleles (functional:non-functional alleles 1:9, 1:99, 1:999). Inheritance bias occurred identically in both sexes with no deposition effect. The fitness impact of the loss of function miR-184$^D$ and non-functional alleles were modelled as an increase in daily mortality relative to wild type (LOF homozygotes, and half impact in heterozygotes), and a mortality probability on each blood meal taken (LOF homozygotes). A single release of 1000 male mosquitoes homozygous for miR-184$^D$ was performed 200 days into the simulation, and simulations were run for a total of 6 years. The 5 years after release were evaluated for their impact on clinical malaria cases in 0–5 year olds, and adult mosquito lifespan. Larval capacity was varied to generate three different vector density scenarios with 15, 35, or 65 infectious bites per person per year. The same transmission seasonality pattern was repeated each year and across transmission settings. Female adults attempt to take a blood meal every day after emergence until they succeed, followed by which they feed every 3 days. The remaining simulation out of 40 (total 1000) not listed in S11–S14 lost the drive allele soon after release and are listed in the supplemental data.

### Reporting summary

Further information on research design is available in the Nature Portfolio Reporting Summary linked to this article.

## Data availability

The data generated in this study have been deposited in the OSF database [https://doi.org/10.17605/OSF.IO/6U7SX]. RNA sequencing data has been deposited in the Sequence Read Archive under accession code PRJNA1210496. Source data are provided with this paper.

## Code availability

The code generated in this study has been deposited in the OSF database [https://doi.org/10.17605/OSF.IO/6U7SX] under an MIT License.

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

## Acknowledgements

The authors would like to thank Philippos Aris Papathanos, Flaia Krsticevic, Jeremy Bouyer, Hamidou Maiga, Irati Aramburu Gonzalez, Umang Bhatia & Valeria Liu. The work was funded by the Bill and Melinda Gates Foundation grants OPP1158151 and INV-058071 to N.W. and G.K.C.

## Author contributions

Conceptualisation: N.W., A.H.; Data curation: S.V., D.V., N.W.; Formal analysis: S.V., D.V., N.W.; Funding acquisition: G.K.C., N.W.; Investigation: G.D.C., P.C., P.S.Y., M.G.I., A.H., AMS; Methodology: G.D.C., PC, PSY; Project administration: G.K.C., N.W.; Resources: G.D.C., P.C., M.G.I., A.H., A.M.S., C.V.U., T.K.; Software: S.V., P.S., N.W.; Modelling: P.S., S.V., N.W.; Supervision: D.V., G.K.C., N.W.; Validation: G.D.C., P.C., A.M.S.; Visualisation: S.V., N.W.; Writing – original draft: N.W.; Writing – review & editing: S.V., G.C.K., N.W.

## Competing interests

The authors declare no competing interests.
