## [Transparent Peer Review file · Nature Communications]

A suppression-modification gene drive for malaria control targeting the ultra-conserved RNA gene mir-184

Corresponding Author: Dr Nikolai Windbichler

Version 0:

Reviewer comments:

Reviewer #1

(Remarks to the Author)

This is an elegant and innovative study that presents a transgenic mosquito -based malaria control approach that has the potential to combine population suppression and modification, and also for the first time involves genetic disruption of a miRNA. Compromised longevity will impact malaria transmission. As the authors outline, targeting miRNAs have several potential benefits in terms of sequence conservation and fitness effects, and the miR-184D locus also seems to be well-suited for over-expression of anti-Plasmodium effectors. The study also addresses the physiological function of miR-184D and provides some hypothetical and reasonable explanations of the KO phenotype. Gene-drive characteristics suggest that miR-184D is suitable for driving it into populations.

The authors say, "(20 and 25 days median lifespan respectively) compared to the wild-type control (median of 32 and 35 days)." It would be useful for the non-Plasmodium sporogonic stage expert reader to elaborate on how this decreased longevity can impact malaria's transmission. The investigators already have the sporozoite infection stage data along with the mortality data in that assay.

The authors say, "We also used miR-184D to co-propagate a non-autonomous antimalarial effector, thus achieving aspects of suppression and modification in these caged populations.", and "Finally, our data indicates that miR-184D was highly effective in propagating the antimalarial effector MM-CP in all 3 cage populations." but this reviewer cannot see any reference to the MM-CP or detailed written results of these assays. Also in the discussion, the authors could elaborate further on this aspect since it is highlighted as a major advantage of their strategy.

The results section is presented as one long continuous text making it difficult to navigate and find descriptions of specific experiments. The authors should introduce headings that clearly define the various sub-studies.

Based on the novelty, quality and public health potential of this study I recommend its publication in Nat Communications.

Reviewer #2

(Remarks to the Author)

Verkuijl et al. have assembled a well-written manuscript thoroughly detailing the design, construction and population genetics of a gene drive DNA construct that accomplishes goals of both population reduction and modification using a Cas9/gRNA disruption of a highly conserved miRNA, {aga-miR-184}. The description of the transgene construct and the characterization of the marker expression, microRNA expression, mosquito fitness and drive characteristics are detailed and show that the drive mechanism is robust. The fitness testing in the drive line demonstrated that knockout of the miRNA had a complex fitness phenotype. These assessments were both thorough and well described; the authors employed a comprehensive set of fitness tests for lab populations and observed a very different set of phenotypes resulting from knockout of miR-184 in *An. gambiae* than was observed in *Drosophila* that included reduction of lifespan, variable impact on female flight ability and mortality following nutrient stress. The removal of GFP with the loxP sites allowed the researchers to distinguish fitness effect from the marker (GFP) expression and miR-184 KO and an examination cause of the mortality associated with blood-feeding was undertaken. Overall, the authors rightly assert that their new approach to a gene drive design has attributes of a suppression drive that targets a highly conserved gene with important fitness genes and effectively

achieves population modification in cage population. In the humble opinion of this reviewer, the most exciting outcome of this work is a proof that the key principle of natural gene drives that even elements with very high fitness loads can be driven into a population is also true of synthetic drives.

I recommend this work for publication with only consideration of the following comments by the authors and a few minor revisions.

Considerations for the authors:

The visual representation of each genotype in the cage population over generations using plots in supplemental figures S7 b and c and S8 b are really effective, and it is very helpful to see the "failed DNA extraction or PCR" included in this, instead of left out.

I understand that there is an intuitive benefit of combining population replacement and suppression strategies to reduce the size of the target population that is ultimately intended to be modified, but is there an additional benefit? Bite reduction benefits of population suppression aren't achieved with this approach since the end result is an intact, but malaria-resistant mosquito population. Do the egg count results in the larger cage experiments results presented here give a nod to a theoretical benefit to driving in a pathogen-resistance allele, then reduce the population over time as a way to eliminate the gene drive from the population and make the population more manageable for other types of elimination programs, should they be used? They project a population that might decrease very slowly over time, but is there any real benefit to this? Or if there is another important utility, can you clarify or expand?

In a similar vein, there is only a hint of suppression in the cage populations, though this is consistent with one generation of drive transgene decline before the drive and non-drive transgene both increase to fixation. In practice, is your drive very different to a standard population modification/replacement drive?

Very Minor Revisions:

In the last paragraph of the background section, the authors state the despite be highly conserved across metazoans, no clear function has emerged for this gene, but this a bit off and I would suggest that you revise this language slightly. Micro RNAs have a well-known function to modulate gene expression, often targeting multiple transcripts, as the authors mention in the discussion. A multifaceted role in gene expression is a clear function and that emerges in this work as a likely function in *An. gambiae* too from the analysis of your transcriptome data and the CLIP-seq data sets as well as from the complex fitness phenotype the authors so thoroughly described.

Some information missing in the methods:

A commonly overlooked part of many transgenic papers are methods used embryo injections. Here the authors have not referenced any source for microinjection methods or given any useable data on injection parameters and success. While these data do not impact the results associated how the final transgenic mosquito lines used, it does limit the reproducibility of this method. It is common for authors to simply reference another paper that vaguely described microinjection methods, but it is this reviewer's opinion that our field would benefit from a more in-depth description of the methods and outcomes of the embryo injections.

In assays for larvae to pupae, please mention the water volume and pan dimensions or larval density, in addition to larvae number. Depending on the volume of water 100 larvae may be very spread out or crowded, which is relevant to comparing fitness between different group's experiments. Similarly cage sizes for adults are relevant.

Reviewer #3

(Remarks to the Author)

This is an interesting article describing the generation of a gene drive targeting a microRNA gene.

Targeting this type of gene is novel and will be interesting to the field. Sequence conservation and functional constraint usually go hand in hand and for microRNAs there are two sources of constraint: base pairing with cognate mRNA targets whose expression (stability or translation potential) is controlled by the microRNA; intramolecular base pairing within the MicroRNA that is required for stem loop formation. For this reason they are potentially attractive as targets and this report is the first describe their targeting by a gene drive. The drive shows good dynamics and the experiments to show their invasive potential in lab populations are well described and well performed. Similarly, the characterisation of fitness effects and life history traits provides useful experimental parameters for modelling how such an element might be expected to perform. That the actual modelling is not done is therefore an oversight that should be addressed. It (the gene drive) does not have to be stellar to warrant inclusion, nor will it diminish the paper if it is not, since it is interesting anyway as first of its kind. The speculation on the function and mode of action of the microRNA is, frankly, too much. I ask myself, would this section stand up to scrutiny alone, if it were written as a paper describing the function and mode of action of miR-184, for which there have been numerous studies in *Drosophila* and where, even with more evidence, conclusions are more guarded and equivocal. My suggestion is that the bulk of this (investigation of miR-184) should be for another paper, with more data down the line, while leaving this article to the dynamics of the novel gene drive. In its pared down form, with the issues below addressed, I would recommend its acceptance.

Overall I found the paper to be well written and easy to follow, which is no mean feat for a topic such as gene drive that can be complex for a non-specialist. That said, there are a few assumptions, such as knowing the make up and nature of the non-autonomous drive elements, and some over-simplifications in the Introduction, that need some attention.

Rather than list these, and others, here I have attached an annotated copy of the manuscript with mark-up in line.

A few to highlight here though are:

the forced emphasis that suggests that suppression and modification approaches, as separate elements, would not ordinarily be synergistic

some of the assertions about measurements of fertility were hard to parse - in some occasions it seemed to be that fertility of carriers was high, if measured only among those that could lay eggs.

with regards to the overspeculation on oxidative stress, aquapores and the rest and "we found that the response to oxidative stress[]was not negatively affected" - you did not measure 'response to oxidative stress' at all, you simply measured lifespan in the presence of paraquat.

If half of the individuals in the starting population were homozygous for the drive element and half were wild type I fail to see how the starting allele frequency is 20%, as stated, rather than 50%

The discussion of the non-drive and non-WT alleles that appear to be drive-induced mutations is interesting and ideally it would have been much more useful, for the purposes of application of the approach, to have investigated these more. Certainly more linearly aligned than the speculation on the miRNA function. I am not suggesting this as a condition of acceptance of the paper; I think there is enough in the drive characterisation to satisfy novelty and general interest. However, I do think there is a lot of speculation and hedging on how these constructs might perform, in given scenarios, and it would be simpler and more satisfying to model that, rather than hand waving in the Discussion.

Some of the 'clear advantages' of this approach, as well as some of its 'clear disadvantages' are over-stated. See annotated text for details

It should probably have been stated earlier in the manuscript, and more explicitly, that the elements experimentally evaluated here are two non-linked drive constructs. Therefore, more similar to the independent activity of a suppressing element and a 'modifying' element, which was alluded to in the intro as being 'not straightforward' and the language implied (incorrectly in my view) that it might not be beneficial or synergistic in most cases. In your case reported here, for the current set up, the probability of synergy, and whether this increases or decreases, is not immediately clear to me (I find myself arguing it both ways) so it will not be clear to most readers either.

Version 1:

Reviewer comments:

Reviewer #1

(Remarks to the Author)

The authors have addressed all comments satisfactory by presenting new data and making appropriate changes in the manuscript.

Reviewer #3

(Remarks to the Author)

The authors have submitted a much improved manuscript. The modelling definitely adds something extra and meaningful, that was not necessarily intuitive, so I am glad they did it.

I still think there is a little too much speculation on mechanism through which these phenotypes are mediated, but the cuts are in the right direction and, in the absence of similar objections from other reviewers, I am not going to insist on further cutting - suffice it so say that the catch all sentence "To fully elucidate the function of miR-184 and its relationship to the phenotypes we observed in mosquitoes, further and more detailed studies are necessary" is doing a lot of lifting here

I am grateful for the tracked changes doc with line numbers, which makes my life easier. Referring to this:

Lines 433 t446 it is not clear to the reader if you are talking about a resistant allele here, the text could be clearer here. I also wonder if the fact that the line can be bred true in the lab might actually be an advantage for another reason, that being that there is a similar selection pressure for resistance in the lab/factory as there is in field, compared with a strain that is constantly outcrossed to WT in the lab/factory yet only faces serious selection pressure for resistance in wild when near elimination. I am not necessarily expecting the authors to include this speculation in the text but I would like an answer to it here.

5.87Mb distance is one thing, but probably should put centimorgans in brackets. From memory this should be about 6cM, so

in those offspring where there was not perfect inheritance of both elements through homing once would expect to see 6% disassociation by homing. Presumably there were just too few overall expected events to say anything meaningful about whether there was deviation from this. I am asking this while thinking about recent articles suggesting that biased chromosome inheritance (meiotic drive) as well as homing can both contribute to biased inheritance of the drive element and whether the authors can exclude that both were at play here. Again, same condition applies - I am not necessarily expecting the authors to include this speculation in the text but I would like an answer to it here.

Version 2:

Reviewer comments:

Reviewer #3

(Remarks to the Author)
thanks for the explanations

(Remarks on code availability)

We thank all reviewers for their considered and positive feedback. We have made changes to the manuscript to address the comments provided. The most substantial change is the inclusion of computational modelling (described after line ~261 Fig6, S9-S15). The author list has been expanded to reflect the modelling work and previous experimental work. We have also made various aesthetic changes to previous figures detailed at the end of this report. We have also deposited the raw data and analysis code.

Below we have formulated point-by-point responses to the reviewer's comments. Reviewers' comments are in black, responses are in blue.

REVIEWER COMMENTS

Reviewer #1 (Remarks to the Author):

This is an elegant and innovative study that presents a transgenic mosquito -based malaria control approach that has the potential to combine population suppression and modification, and also for the first time involves genetic disruption of a miRNA. Compromised longevity will impact malaria transmission. As the authors outline, targeting miRNAs have several potential benefits in terms of sequence conservation and fitness effects, and the miR-184D locus also seems to be well-suited for over-expression of anti-Plasmodium effectors. The study also addresses the physiological function of miR-184D and provides some hypothetical and reasonable explanations of the KO phenotype. Gene-drive characteristics suggest that miR-184D is suitable for driving it into populations.

The authors say, "(20 and 25 days median lifespan respectively) compared to the wild-type control (median of 32 and 35 days)." It would be useful for the non-Plasmodium sporogonic stage expert reader to elaborate on how this decreased longevity can impact malaria transmission. The investigators already have the sporozoite infection stage data along with the mortality data in that assay.

The addition of the new gene drive and malaria model will hopefully address the questions relating to the implications of a reduced lifespan on malaria transmission.

Lines ~261

The authors say, "We also used miR-184D to co-propagate a non-autonomous antimalarial effector, thus achieving aspects of suppression and modification in these caged populations.", and "Finally, our data indicates that miR-184D was highly effective in propagating the antimalarial effector MM-CP in all 3 cage populations." but this reviewer cannot see any reference to the MM-CP or detailed written results of these assays. Also in the discussion, the authors could elaborate further on this aspect since it is highlighted as a major advantage of their strategy.

We now include a brief description and more explicit reference to the previously published MM-CP modification.

Lines ~232

The results section is presented as one long continuous text making it difficult to navigate and find descriptions of specific experiments. The authors should introduce headings that clearly define the various sub-studies.

We have added the following headings to the revised results section:

- Gene drive construct and transmission efficiency at the mir-184 locus
- miR-184^D impacts mosquito lifespan and flight ability
- miR-184^D mosquitoes experience blood meal-associated stress
- Transcriptional perturbation of solute transport in miR-184^D midguts
- miR-184^D population invasion experiments
- Gene drive and malaria transmission model

Based on the novelty, quality and public health potential of this study I recommend its publication in Nat Communications.

Reviewer #2 (Remarks to the Author):

Verkuijl et al. have assembled a well-written manuscript thoroughly detailing the design, construction and population genetics of a gene drive DNA construct that accomplishes goals of both population reduction and modification using a Cas9/gRNA disruption of a highly conserved miRNA, {aga-miR-184}. The description of the transgene construct and the characterization of the marker expression, microRNA expression, mosquito fitness and drive characteristics are detailed and show that the drive mechanism is robust. The fitness testing in the drive line demonstrated that knockout of the miRNA had a complex fitness phenotype. These assessments were both thorough and well described; the authors employed a comprehensive set of fitness tests for lab populations and observed a very different set of phenotypes resulting from knockout of miR-184 in *An. gambiae* than was observed in *Drosophila* that included reduction of lifespan, variable impact on female flight ability and mortality following nutrient stress. The removal of GFP with the loxP sites allowed the researchers to distinguish fitness effect from the marker (GFP) expression and miR-184 KO and an examination cause of the mortality associated with blood-feeding was undertaken. Overall, the authors rightly assert that their new approach to a gene drive design has attributes of a suppression drive that targets a highly conserved gene with important fitness genes and effectively achieves population modification in cage population. In the humble opinion of this reviewer, the most exciting outcome of this work is a proof that the key principle of natural gene drives that even elements with very high fitness loads can be driven into a population is also true of synthetic drives.

I recommend this work for publication with only consideration of the following comments by the authors and a few minor revisions.

Considerations for the authors:

The visual representation of each genotype in the cage population over generations using plots in supplemental figures S7 b and c and S8 b are really effective, and it is very helpful to see the “failed DNA extraction or PCR” included in this, instead of left out.

We appreciate the positive feedback.

I understand that there is an intuitive benefit of combining population replacement and suppression strategies to reduce the size of the target population that is ultimately intended to be modified, but is there an additional benefit? Bite reduction benefits of population suppression aren't achieved with this approach since the end result is an intact, but malaria-resistant mosquito population. Do the egg count results in the larger cage experiments results presented here give a nod to a theoretical benefit to driving in a pathogen-resistance allele, then reduce the population over time as a way to eliminate the gene drive from the population and make the population more manageable for other types of elimination programs, should they be used? They project a population that might be decrease very slowly over time, but is there any real benefit to this? Or if there is another important utility, can you clarify or expand?

We find that the miR-184 drive element is able to efficiently invade and maintain itself at a high frequency despite imposing a significant fitness cost in the lab. In the field, various other factors may come into place for example density dependence. Still, if the imposed fitness reduction can be maintained over long periods, one could expect the population size to decrease to a new lower equilibrium population size. In areas and seasons where infectious bites are not excessively saturating, we would expect this to decrease the malaria burden.

If a perfectly protective anti-malarial effector is identified, the over-saturation of infectious mosquito bites would indeed not matter. However, to date, the effector candidates being explored have only an intermediate protective effect. As such, intermediate anti-malaria protection could be enhanced with miR-184^D suppression. Of note is that the reduced lifespan of females, like the biting ability in your example, also directly impacts their ability to transmit malaria independent of a change to the population size.

The question the reviewer raises is complex. See also reviewer 3. To make a first stab at addressing all these aspects we now have included modelling of the propagation and expected impact of gene drives similar to miR-184^D on malaria outcomes which expands on the points raised above.

Line ~261

In a similar vein, there is only a hint of suppression in the cage populations, though this is consistently with one generation of drive transgene decline before the drive and non-drive transgene both increase to fixation. In practice, is your drive very different to a standard population modification/replacement drive?

Our drive is different in that we demonstrate it can spread to and stay at high-frequencies despite imposing substantial fitness costs. Comparable experiments are e.g. the *Anopheles* replacement drive targeting the *kmo* gene failed to maintain high frequencies despite having similarly high inheritance rates due to its fitness cost¹. More recent replacement drives from the same group have aimed to achieve high frequencies by incorporating a rescue allele² or targeting genes with milder fitness costs^{3,4}. This allows them to perform similar to our drive in their cage trials, however, without the corresponding constant suppression benefit our drive confers. In the case of MM-CP, it can maintain a high frequency because its fitness costs are moderate⁵. We hypothesised that the miR-184^D modification can maintain itself at high frequency because (dominant) functional resistance did not emerge.

A large caveat is, as we mention in the text, that the true fitness cost of the loss of miR-184 in the wild is unknown and can't be inferred from limited fitness assays we are able to perform in the lab. It could be similar or it could be substantially higher in which case the drive would behave closer to a suppression drive and the selection pressure for resistance alleles might also be elevated accordingly.

Very Minor Revisions:

In the last paragraph of the background section, the authors state the despite be highly conserved across metazoans, no clear function has emerged for this gene, but this a bit off and I would suggest that you revise this language slightly. Micro RNAs have a well-known function to modulate gene expression, often targeting multiple transcripts, as the authors mention in the discussion. A multifaceted role in gene expression is a clear function and that emerges in this work as a likely function in *An. gambiae* too from the analysis of your transcriptome data and the CLIP-seq data sets as well as from the complex fitness phenotype the authors so thoroughly described.

We have adjusted this section to instead make the point that no clear *common* function has emerged. (Line ~90)

Some information missing in the methods:

A commonly overlooked part of many transgenic papers are methods used embryo injections. Here the authors have not referenced any source for microinjection methods or given any useable data on injection parameters and success. While these data do not impact the results associated how the final transgenic mosquito lines used, it does limit the reproducibility of this method. It is common for authors to simply reference another paper that vaguely described microinjection methods, but it is this reviewer's opinion that our field would benefit from a more in-depth description of the methods and outcomes of the embryo injections.

We have expanded the description of the microinjections in the methods and cited a JoVE video detailing the equivalent injection procedure we followed. (Line ~440)

In assays for larvae to pupae, please mention the water volume and pan dimensions or larval density, in addition to larvae number. Depending on the volume of water 100 larvae may be very spread out or crowded, which is relevant to comparing fitness between different group's experiments. Similarly, cage sizes for adults are relevant.

We have updated the methods with the following statement:

Mosquito aquatic stages were kept in 25² cm trays with ~500ml of water and adults were housed in 17.5³ cm cages, or 25³ cm in the case of the cage trial.
(Line ~460)

Reviewer #3 (Remarks to the Author):

This is an interesting article describing the generation of a gene drive targeting a microRNA gene.

Targeting this type of gene is novel and will be interesting to the field. Sequence conservation and functional constraint usually go hand in hand and for microRNAs there are two sources of constraint: base pairing with cognate mRNA targets whose expression (stability or translation potential) is controlled by the microRNA; intramolecular base pairing within the MicroRNA that is required for stem loop formation. For this reason they are potentially attractive as targets and this report is the first describe their targeting by a gene drive. The drive shows good dynamics and the experiments to show their invasive potential in lab populations are well described and well performed. Similarly, the characterisation of fitness effects and life history traits provides useful experimental parameters for modelling how such an element might be expected to perform. That the actual modelling is not done is therefore an oversight that should be addressed. It (the gene drive) does not have to be stellar to warrant inclusion, nor will it diminish the paper if it is not, since it is interesting anyway as first of its kind.

We have now included modelling work as suggested by the reviewer where we explore how the phenotypes induced by a drive like miR-184^D would affect its propagation and modify malaria transmission. The reviewer was correct to push us to do this, as the results are interesting.

The speculation on the function and mode of action of the microRNA is, frankly, too much. I ask myself, would this section stand up to scrutiny alone, if it were written as a paper describing the function and mode of action of miR-184, for which there have been numerous studies in *Drosophila* and where, even with more evidence, conclusions are more guarded and equivocal. My suggestion is that the bulk of this (investigation of miR-184) should be for another paper, with more data down the line, while leaving this article to the dynamics of the novel gene drive. In its pared down form, with the issues below addressed, I would recommend its acceptance.

We have substantially reduced the length of some of these sections and made a better effort to indicate what conclusions are solidly underpinned by data and when we are treading on the more speculative side of the function of miR-184.

However, while the reviewer is correct to point out that there are already some functional studies on miR-184 in the literature including *Drosophila* work, many of them were obtained using antagomirs or larger deletions of the miR-184 genomic region. Our study provides, as we show, a clean KO of miR-184 and a well-powered RNA-seq dataset in an organism with a different metabolic regime (hematophagous lifestyle with miR-184 being highly expressed in the midgut). We believe therefore that venturing into the biology remains worthwhile here.

Overall I found the paper to be well written and easy to follow, which is no mean feat for a topic such as gene drive that can be complex for a non-specialist. That said, there are a few assumptions, such as knowing the make up and nature of the non-autonomous drive elements, and some over-simplifications in the Introduction, that need some attention.

Rather than list these, and others, here I have attached an annotated copy of the manuscript with mark-up in line.

We thank the reviewer for their detailed reading of the text. We have adjusted the manuscript accordingly. These changes are reflected in the track-changes document.

A few to highlight here though are:

the forced emphasis that suggests that suppression and modification approaches, as separate elements, would not ordinarily be synergistic

We have now made explicit that this refers to full suppression drives, see also the last point below. Hopefully, this addresses the confusion.

Line ~50, Line ~70

some of the assertions about measurements of fertility were hard to parse - in some occasions it seemed to be that fertility of carriers was high, if measured only among those that could lay eggs.

We have revised the text for clarity. Line ~120

with regards to the overspeculation on oxidative stress, aquapores and the rest and "we found that the response to oxidative stress [] was not negatively affected" - you did not measure 'response to oxidative stress' at all, you simply measured lifespan in the presence of paraquat.

We have adjusted the language of this section of the results to a more careful tone. Having said that, oxidative stress is known to limit the lifespan of multicellular organisms^{6,7} and paraquat is commonly used to induce oxidative stress in order to measure this⁸. Therefore we are not making new claims.

Line ~165

If half of the individuals in the starting population were homozygous for the drive element and half were wild type I fail to see how the starting allele frequency is 20%, as stated, rather than 50%

We have adjusted the text for clarity. The half-half here referred to the ratio between males and females but we agree this was confusing.

Line ~225

The discussion of the non-drive and non-WT alleles that appear to be drive-induced mutations is interesting and ideally it would have been much more useful, for the purposes of application of the approach, to have investigated these more. Certainly more linearly aligned than the speculation on the miRNA function. I am not suggesting this as a condition of acceptance of the paper; I think there is enough in the drive characterisation to satisfy novelty and general interest. However, I do think there is a lot of speculation and hedging on how these constructs might perform, in given scenarios, and it would be simpler and more satisfying to model that, rather than hand waving in the Discussion.

We have now included computational modelling in the revised manuscript.

Some of the 'clear advantages' of this approach, as well as some of its 'clear disadvantages' are over-stated. See annotated text for details

We have adjusted the text accordingly.

Line ~50-70 and Line ~354

It should probably have been stated earlier in the manuscript, and more explicitly, that the elements experimentally evaluated here are two non-linked drive constructs.

With regards to the elements being unlinked, we now more clearly state that these are separate insertions. Of note, CP and mir-184 are both located on the right arm of chromosome 3 (separated by almost 6 Mb). As a side note, we do not think this affects the performance of the drives, as with near-perfect inheritance rates, the two gene drive elements remain linked regardless of recombination distance.

Therefore, more similar to the independent activity of a suppressing element and a 'modifying' element, which was alluded to in the intro as being 'not straightforward' and the language implied (incorrectly in my view) that it might not be beneficial or synergistic in most cases. In your case reported here, for the current set up, the probability of synergy, and whether this increases or decreases, is not immediately clear to me (I find myself arguing it both ways) so it will not be clear to most readers either.

The initial argument in the introduction about synergy between a suppressive drive and a modification drive was focussed on perfect suppression that seeks to completely abolish mosquito reproduction for example. If such an element is successful in eliminating a population there is no space for additional synergy. If it is not successful, this can be due to various reasons (population structure, resistance etc.) leading to a complex interaction with a modifying drive. For example, if a chasing dynamic arises in the population, it is not clear that a modification drive would not perform better spreading through an original unaltered and fully-connected WT population. In that case, combined suppression/modification may lead to lower reduction in malaria than the release of a modification drive alone. This is what was meant with synergy not being straightforward.

The reviewer has a fair point, in the sense that there is no broad modelling-based literature that fully explores the interaction between suppression and modification drives and much more such work needs to be done to better understand these complexities. We have softened the language to reflect this.

In our case, the miR-184^D drive we describe already combines aspects of suppression and modification as it is. An example of the suppressive effect it exerts is the mortality following the bloodmeal leading to lower reproductive output. However, the drive fixes/plateaus in the population and affects mosquito lifespan (and hence vectorial capacity) which is more akin to a modification drive. The inclusion of the model hopefully now better describes the resulting net effect on malaria from the combined action of modification and suppression aspects of this drive on its own.

Because we are left with a reproducing mosquito population that expresses Cas9, the co-propagation of non-autonomous effector traits becomes a further possibility and this is what we have shown experimentally by adding the MM-CP non-autonomous effector trait.

Here additional synergy arises, as effectors like MM-CP can directly target parasite development in transgenic mosquitoes.

We have now included the arguments above in the paper for hopefully improved clarity.

Minor changes:

- Various figure x-axis labels have been realigned.
- 1a: The gRNA binding site has been changed to purple matching the cage trial.
- The colour legend of 2b had a female symbol instead of a male for the matching blue and red colours. This has been corrected.
- Included missing * in figure legend of S2b.
- Realigned text in 4a for clarity.
- Changed drive symbol from D, to d in S3.

Bibliography

1. Pham, T. B. *et al.* Experimental population modification of the malaria vector mosquito, *Anopheles stephensi*. *PLOS Genet.* **15**, (2019).
2. Adolfi, A. *et al.* Efficient population modification gene-drive rescue system in the malaria mosquito *Anopheles stephensi*. *Nat. Commun.* **11**, 5553 (2020).
3. Carballar-Lejarazú, R. *et al.* Next-generation gene drive for population modification of the malaria vector mosquito, *Anopheles gambiae*. *Proc. Natl. Acad. Sci. U. S. A.* **117**, 22805–22814 (2020).
4. Carballar-Lejarazú, R. *et al.* Dual effector population modification gene-drive strains of the African malaria mosquitoes, *Anopheles gambiae* and *Anopheles coluzzii*. *Proc. Natl. Acad. Sci.* **120**, e2221118120 (2023).
5. Hoermann, A. *et al.* Gene drive mosquitoes can aid malaria elimination by retarding *Plasmodium* sporogonic development. (2022) doi:10.1101/2022.02.15.480588.
6. Guarente, L. & Kenyon, C. Genetic pathways that regulate ageing in model organisms. *Nature* **408**, 255–262 (2000).
7. Finkel, T. & Holbrook, N. J. Oxidants, oxidative stress and the biology of ageing. *Nature* **408**, 239–247 (2000).
8. Girardot, F., Monnier, V. & Tricoire, H. Genome wide analysis of common and specific stress responses in adult *Drosophila melanogaster*. *BMC Genomics* **5**, 74 (2004).

We thank all reviewers for their considered and positive feedback.

Below we have formulated point-by-point responses to the reviewer's comments. Reviewers' comments are in black, responses are in blue.

REVIEWER COMMENTS

Reviewer #1 (Remarks to the Author):

The authors have addressed all comments satisfactory by presenting new data and making appropriate changes in the manuscript.

We thank the reviewer.

Reviewer #3

The authors have submitted a much improved manuscript. The modelling definitely adds something extra and meaningful, that was not necessarily intuitive, so I am glad they did it. I still think there is a little too much speculation on mechanism through which these phenotypes are mediated, but the cuts are in the right direction and, in the absence of similar objections from other reviewers, I am not going to insist on further cutting - suffice it so say that the catch all sentence "To fully elucidate the function of miR-184 and its relationship to the phenotypes we observed in mosquitoes, further and more detailed studies are necessary" is doing a lot of lifting here

I am grateful for the tracked changes doc with line numbers, which makes my life easier. Referring to this:

Lines 433 t446 it is not clear to the reader if you are talking about a resistant allele here, the text could be clearer here.

We have changed this to "cleavage-resistant miR-184 alleles" to increase clarity.

I also wonder if the fact that the line can be bred true in the lab might actually be an advantage for another reason, that being that there is a similar selection pressure for resistance in the lab/factory as there is in field, compared with a strain that is constantly outcrossed to WT in the lab/factory yet only faces serious selection pressure for resistance in wild when near elimination. I am not necessarily expecting the authors to include this speculation in the text but I would like an answer to it here.

We think that the resistance implications of a true breeding strain compared to a gene drive strain that must constantly be outcrossed are potentially complex.

We should distinguish between the selection for gene drive resistance and resistance against / mitigation of the phenotype induced by the drive. The former cannot arise via constant backcrossing/screening and is equally unlikely to be selected for during the maintenance of an already homozygous true-breeding strain.

The latter could occur as a compensatory mutation within or linked to the drive element and could e.g. be selected for if the suppression drive being backcrossed shows heterozygous

fitness costs. It could also arise as an adaptive, compensatory mutation in the true breeding strain. This depends on the degree to which the same (magnitude of) selective pressures would act in the lab and the field and if some of these effects can be suppressed. For example, in the case of the miR-184^D drive we know that the constant provision of sugar could mitigate the mortality associated with blood feeding and therefore reduce the selective pressure for compensatory mutations.

A true breeding strain is likely to be less genetically diverse than the WT colony a backcrossing drive strain would be outcrossed to. As such, while the selection pressure might be higher in the true breeding strain, the diversity of alleles on which the selection pressure can act on is smaller. Lastly, an already homozygous true breeding line may become more fit if it develops spontaneous mutations in the nuclease genes (e.g., Cas9).

We do not think this is an exhaustive list, but it highlights the potential complex implications for resistance that we cannot do justice in the limited space we have in the manuscript.

5.87Mb distance is one thing, but probably should put centimorgans in brackets. From memory this should be about 6cM, so in those offspring where there was not perfect inheritance of both elements through homing once would expect to see 6% disassociation by homing. Presumably there were just too few overall expected events to say anything meaningful about whether there was deviation from this. I am asking this while thinking about recent articles suggesting that biased chromosome inheritance (meiotic drive) as well as homing can both contribute to biased inheritance of the drive element and whether the authors can exclude that both were at play here. Again, same condition applies - I am not necessarily expecting the authors to include this speculation in the text but I would like an answer to it here.

Due to the high expected inheritance bias of each element in our cage experiments, the informative non-drive inheriting progeny was rare and the 92 progeny we genotyped are not sufficient to draw a meaningful conclusion with regards to this issue. Moreover, analysis of the offspring of the expected double-hets that emerge in the cage trial is further obscured by the potential for intercrosses in addition to the more informative outcrosses to WT. We did not perform isolated crosses with miR-184^D and MM-CP to specifically address this.

One thing we can say is that when the first double-hets are created in the cage trial in generation 1 the miR-184^D and MM-CP transgenes will necessarily have been located on opposite chromosomes (homozygous individuals of each line were used to initiate our cage trial). If meiotic drive was the predominant mechanism of inheritance bias we expect this would have severely hampered the performance of the drive (both homologs would be cut simultaneously), which does not match the observed high performance in the cage trials.

With regards to genetic distance: There is no direct interconversion between genetic (cM) and genomic (Mb) distances and we would need to establish the genetic distance experimentally for the miR-184 and CP loci. Measuring background recombination rate is best done by creating double hets with different markers at each site and outcrossing to WT. However, we only have insertions in the form of gene drive elements, and could not estimate the background rate accurately enough (because of gene drive) to serve as a robust control without making additional transgenic lines lacking drive.